# Prior infection with unrelated neurotropic virus exacerbates influenza disease and impairs lung T cell responses

Isabelle Jia-Hui Foo[1,2], Brendon Y. Chua [1], E. Bridie Clemens[1], So Young Chang[1], Xiaoxiao Jia[1], Hayley A. McQuilten [1], Ashley Huey Yiing Yap[1], Aira F. Cabug[1], Mitra Ashayeripanah[1], Hamish E. G. McWilliam [1], Jose A. Villadangos [1,3], Maximilien Evrard[1], Laura K. Mackay [1], Linda M. Wakim[1], John K. Fazakerley [1,2,4], Katherine Kedzierska [1,4] ✉ & Lukasz Kedzierski [1,4] ✉

Immunity to infectious diseases is predominantly studied by measuring immune responses towards a single pathogen, although co-infections are common. In-depth mechanisms on how co-infections impact anti-viral immunity are lacking, but are highly relevant to treatment and prevention. We established a mouse model of co-infection with unrelated viruses, influenza A (IAV) and Semliki Forest virus (SFV), causing disease in different organ systems. SFV infection eight days before IAV infection results in prolonged IAV replication, elevated cytokine/chemokine levels and exacerbated lung pathology. This is associated with impaired lung IAV-specific CD8[+] T cell responses, stemming from suboptimal CD8[+] T cell activation and proliferation in draining lymph nodes, and dendritic cell paralysis. Prior SFV infection leads to increased blood brain barrier permeability and presence of IAV RNA in brain, associated with increased trafficking of IAV-specific CD8[+] T cells and establishment of long-term tissue-resident memory. Relative to lung IAV-specific CD8[+] T cells, brain memory IAV-specific CD8[+] T cells have increased TCR repertoire diversity within immunodominant $D^bNP_{366}^+CD8^+$ and $D^bPA_{224}^+CD8^+$ responses, featuring suboptimal TCR clonotypes. Overall, our study demonstrates that infection with an unrelated neurotropic virus perturbs IAV-specific immune responses and exacerbates IAV disease. Our work provides key insights into therapy and vaccine regimens directed against unrelated pathogens.

Immunity to infectious diseases is generally studied in isolation by measuring immune responses towards a single pathogen. The reality is that we can harbour more than one infection at any given time, in sequence or concurrently, which can substantially affect immune responses to subsequent infection or vaccination. Co-infections with two or more pathogens can occur simultaneously or sequentially, by the same or different routes of transmission. In general, co-infections have a negative effect on human health leading to exacerbation of disease[1].

[1]Department of Microbiology and Immunology, The University of Melbourne, at the Peter Doherty Institute for Infection and Immunity, Melbourne VIC 3000, Australia. [2]Department of Veterinary Biosciences, Faculty of Science, The University of Melbourne, Melbourne VIC 3000, Australia. [3]Department of Biochemistry and Pharmacology; Bio21 Molecular Science and Biotechnology Institute, The University of Melbourne, Parkville VIC 3010, Australia. [4]These authors contributed equally: John K. Fazakerley, Katherine Kedzierska, Lukasz Kedzierski. ✉e-mail: kkedz@unimelb.edu.au; lukaszk@unimelb.edu.au

Viral co-infections can increase the variability of anti-viral immune responses, reduce protection and enhance immunopathology[2]. However, differences in immune mechanisms between single infection and co-infection, and the universal burden of viral co-infection on human health remain elusive.

Influenza is a global health concern, causing annual epidemics and sporadic pandemics[3]. The past 50 years have seen the emergence/re-emergence of epidemic arthropod-borne viral (arboviral) diseases on a global scale, with several arboviruses capable of infecting the central nervous system (CNS)[4]. Given the widespread prevalence of both influenza and arboviruses, the likelihood of co-infection with these viruses is high. There are limited data on whether co-infection of influenza A (IAV) with non-respiratory viruses affects disease outcome. During the 2009 H1N1 pandemic, influenza-dengue (DENV) co-infections were reported[5,6], and identified as a risk factor for severe disease[7]. Previous co-infection studies involving influenza and herpes virus, reported that in a mouse model, co-infection with these viruses results in alleviated respiratory symptoms due to influenza infection[8].

As most studies investigate adaptive immune responses in a single pathogen model, immune responses during co-infections are under-studied. Our knowledge of immune responses during viral co-infections is limited and understanding of interactions within the network of site-specific and systemic immune processes is lacking[9]. Of widespread relevance is the effect that one virus infection has on generation of immune responses to a second virus infection during both acute and memory phases. Importantly, in-depth data on immune responses to respiratory viruses during co-infection with unrelated pathogens are lacking.

Here, we comprehensively defined how IAV-specific anti-viral immune responses are modulated by prior infection with the unrelated neurotropic arbovirus Semliki Forest virus (SFV). We established a mouse model to study sequential infection with two unrelated viruses in two key tissues, the lung and the brain. Our data show that sequential-infection (SFV→IAV) results in trafficking of IAV-specific T cells from the lung to the brain, elevated cytokine levels and altered viral replication kinetics in the lungs. Prior SFV infection leads to perturbed proliferation of IAV-specific CD8+ T cells, and ultimately results in impaired immunological memory formation and recall responses. Detailed analysis of TCR repertoires within $D^bNP_{366}{}^+CD8^+$ and $D^bPA_{224}{}^+CD8^+$ clonotypes in the lung and brain revealed a significant increase in clonotype diversity during SFV→IAV infection, indicating a more heterogenous response. Collectively, our study indicates that SFV→IAV sequentially infected animals have more severe respiratory disease and perturbed immune responses than mice infected with IAV alone.

## Results

### Prior infection with non-respiratory SFV leads to exacerbated influenza disease

To define the impact of prior infection with an unrelated non-respiratory virus on influenza virus infection, we established a mouse model of sequential viral co-infection using two well-characterised animal disease models: (i) neurotropic arboviral SFV infection, and (ii) respiratory IAV infection. SFV is a well-studied tractable model of experimental viral encephalitis[10]. In mice, the virus is 100% neu-roinvasive following intraperitoneal injection and does not require intracranial inoculation as is the case with many other encephalitic viruses. SFV easily infects common strains of laboratory mice, and rodents may be its natural host. This negates the need for IFNAR-/- mice used to study some other neurotropic viruses, for example flaviviruses[11]. SFV infection in mice has a considerable body of supporting previous research and provides a well-regarded experimental system to study the pathogenesis of viral encephalitis. Adult C57BL/6 mice were infected with either $10^4$ pfu A/HKx31 (x31) IAV alone via the intranasal (i.n.) route, leading to localised respiratory infection; or

$5×10^3$ pfu A7(74) SFV via the intraperitoneal (i.p.) route, which establishes systemic infection prior to neuroinvasion[12]. In a third group, SFV infection was followed by i.n. IAV infection on day 8 post-SFV (SFV→IAV sequential infection) (Fig. 1a). Day 8 was chosen as adaptive immune responses are at their peak and infectious SFV is generally cleared from the brain around this time[13,14].

Viral, inflammatory and immunological analyses were performed following either IAV alone or sequential SFV→IAV infection during acute and long-term time-points following IAV infection. Mice were monitored for IAV replication and inflammation in the infected lungs. Analysis of IAV lung viral titres by a plaque assay[15] clearly demonstrated that mice exposed to SFV prior to IAV had distorted kinetics of IAV viral replication in the lungs (Fig. 1b). Initially on day (d) 3 after IAV infection, SFV→IAV animals had significantly lower lung viral titres compared to IAV-only mice, but by d7 their lung viral titres were higher. We also found higher levels of lung cytokines/chemokines in the SFV→IAV group relative to IAV-only group at d7 after IAV infection (Fig. 1c), suggesting exacerbated and prolonged IAV disease. SFV-only infection elicited negligible cytokine responses in lung or serum compared to other experimental groups. Elevated inflammatory mediators included IL-6, IL-1β, IFNγ and MCP-1, a signature associated with severe and fatal influenza disease in humans[16,17]. IFNγ levels were also significantly upregulated in the serum of SFV→IAV animals (Fig. 1c). Thus, SFV infection predisposes to elevated cytokine storm following subsequent IAV infection, both at the site of infection and systemically. It also leads to higher IAV titres in the lungs at a later stage of infection.

In agreement with the lung viral load and cytokine data, histopathological analysis of lung tissue from IAV alone and SFV→IAV mice revealed more extensive histopathological changes in the sequentially infected group (Fig. 1d). These included multifocal areas of alveolar interstitial lymphocytic and macrophage infiltration, numerous alveolar macrophages, type II pneumocyte hyperplasia, severe lymphocytic vasculitis and perivascular cuffing, and bronchial/bronchiolar epithelial degeneration/necrosis and hyperplasia accompanied by multifocal haemorrhage (Fig. 1d). Overall, lung damage in SFV→IAV group was more extensive than in the IAV alone group (Fig. 1e). SFV-only infection did not cause detectable histopathological changes in the lungs (Figs. 1d and e).

To understand immune responses during sequential SFV→IAV infection, we performed a broad analysis of myeloid and B cell compartments in lungs, spleen and brain on d7 after IAV infection or SFV infection (Fig. 1f). We found similar numbers of B cells (B220+CD19+) and antibody-secreting cells (ASC, IgD-B220loCD138+) across all tissues examined. Similarly, comparable numbers of neutrophils (CD11b+Ly6G+), macrophages (CD11b+CD64+), DCs (CD11c+) and inflammatory monocytes (CD11b+Ly6C+) were detected across the experimental groups. However, it is plausible that although B cell responses did not differ quantitatively, they could differ qualitatively. B cells from co-infected mice could have different activation status, produced antibodies of lower affinity resulting in impaired affinity maturation, and there could have been differences in the rate of class-switching and numbers of neutralising IgM and IgG antibodies, which could further affect seroconversion and seroprotection rates.

### SFV→IAV sequential infection perturbs activation and trafficking of IAV-specific T cells

To investigate mechanisms underpinning exacerbated IAV disease severity in SFV→IAV mice, we performed in-depth immune analyses of T cell responses in lungs, brain and spleen. We examined total numbers of pan CD4+ and CD8+ T cells (Fig. S1b, c) and effector T cells. The striking and unexpected observation was that at d7, SFV→IAV animals had significantly increased numbers of activated effector CD62LloCD44hi CD4+ (Fig. 1f) and CD62LloCD44hi CD8+ T cells in the brain (Fig. 2a), although no differences were detected in the lungs or the spleen. To determine the specificity of the effector CD8+ T cells in

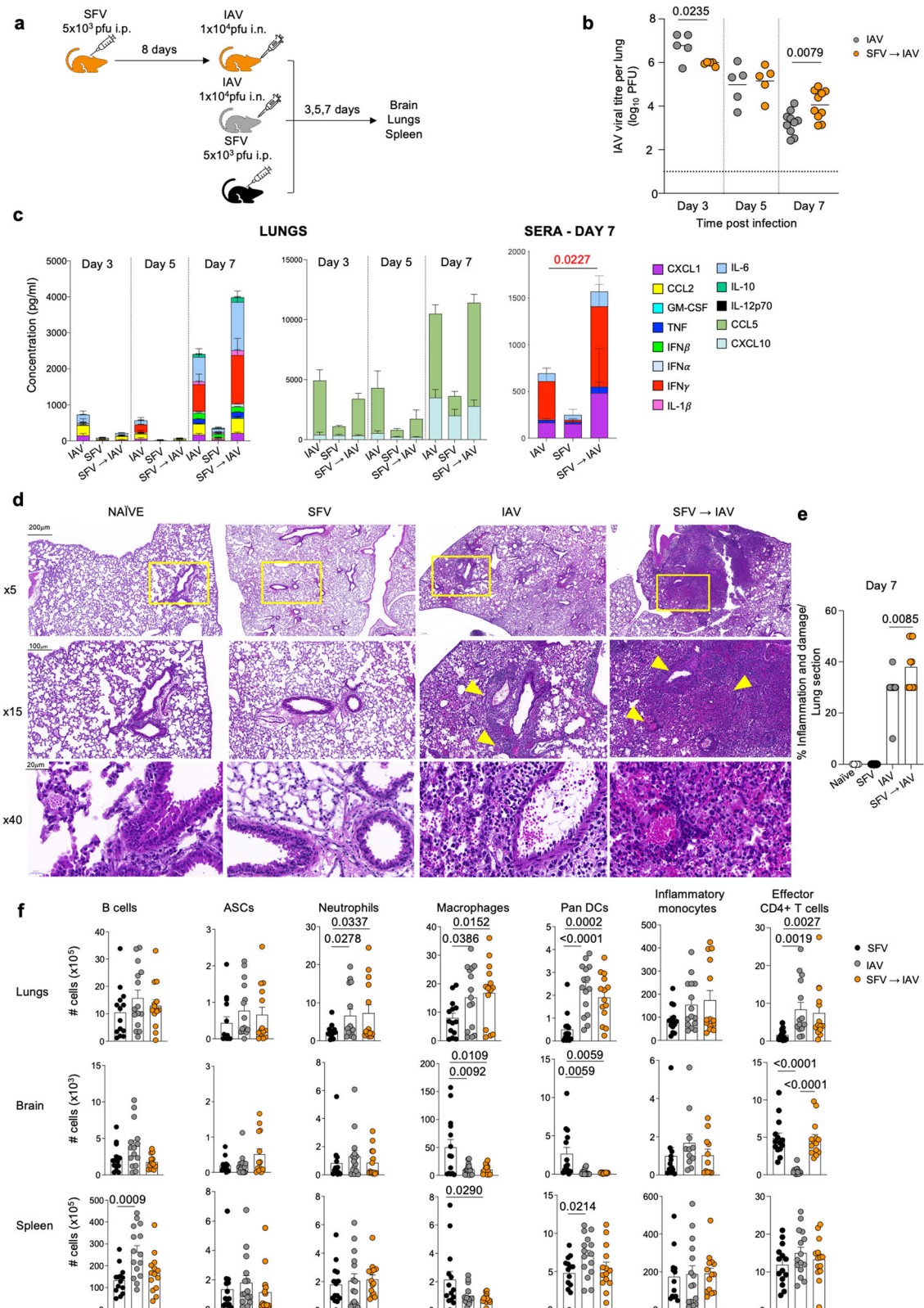

the brain, we defined influenza-specific CD8+ T cells with peptide-MHC-I tetramers ($D^bNP_{366}$ and $D^bPA_{224}$; two immunodominant CD8+ T cell IAV epitopes in B6 mice[18]). In SFV→IAV mice, IAV-specific $D^bNP_{366}^+CD8^+$ and $D^bPA_{224}^+CD8^+$ T cells trafficked to the brain and were detected in 11/15 9/15 of mice, respectively (Fig. 2b). Strikingly, SFV→IAV-infected animals had significantly reduced numbers of $D^bNP_{366}^+CD8^+$ and $D^bPA_{224}^+CD8^+$ T cells in the lung and spleen (Fig. 2b),

demonstrating that kinetics and anatomical distribution (trafficking) of IAV-specific immune responses were markedly perturbed by prior SFV infection. Conversely, of the very few CD8+ T cells infiltrating the brain in the IAV-only group, the majority (95.3%) were IAV tetramer-negative. We could minimally detect $D^bNP_{366}^+CD8^+$ T cells in 4 out of 15 mice and $D^bPA_{224}^+CD8^+$ T cells in 7/15 mice. While the numbers of activated CD4+ T cells mirrored those of activated CD8+ T cells,

**Fig. 1 | SFV→IAV sequential infection exacerbates respiratory disease. a** Mice were infected i.p. with $5 \times 10^3$ pfu A7(74) SFV followed 8 days later with $10^4$ pfu A/HKx31 (i.n.). Brain, lungs and spleens were harvested at various days post-infection (dpi). **b** Lung IAV viral titres were determined by a plaque assay on MDCK cells. Each symbol denotes an individual mouse (d3 IAV $n = 5$, SFV→IAV $n = 5$; d5 IAV $n = 5$, SFV→IAV $n = 5$; d7 IAV $n = 10$, SFV→IAV $n = 10$). **c** Lung homogenates and sera were assayed by LEGENDplex to determine the cumulative amount of total cytokines and chemokines present and the amounts of each cytokine. (SFV→IAV $n = 5$ (d3 and d5), $n = 10$ (d7), IAV $n = 5$ (d3 and d5), $n = 10$ (d7), data from 2 independent experiments are shown, mean values plotted, error bar represents SEM). **d** Histopathological changes in the lungs of SFV, IAV, and SFV → IAV infected mice on 7 dpi. Lungs from naïve mice were included as control. Lung sections were stained with HE;

representative images are shown. Bar = 200 μm for ×5; 100 μm for ×15; and 20 μm for ×40 magnification. Yellow arrows point to area of inflammation and damage. **e** Quantification of the extent of inflammation and damage in SFV, IAV, and SFV→IAV groups (SFV $n = 3$, IAV $n = 3$, SFV→IAV $n = 2$, mean values plotted, error bar represents SEM). **f** Absolute numbers of B cells (B220$^+$CD19$^+$), antibody secreting cells (ASC, IgD B220$^{lo}$CD138$^+$), neutrophils (CD11b$^+$Ly6G$^+$), macrophages (CD11b$^+$CD64$^+$) DCs (CD11c$^+$), inflammatory monocytes (CD11b$^+$Ly6C$^+$) and effector CD4$^+$ T cells (CD62L$^{lo}$CD44$^{hi}$CD4$^+$) across different anatomical sites (SFV $n = 14$, IAV $n = 15$, SFV→IAV $n = 14$, error bar represents SEM). Gating strategy is shown in Supplementary Fig. S5a–c. Significance was determined by unpaired two-tailed Student's t-test. Source data are provided as a Source Data file.

numbers of IAV-specific CD4$^+$ T cells could not be investigated due to poor performance of class II tetramers.

Taking advantage of our recently identified immunodominant SFV-specific CD8$^+$ T cell epitope (K$^b$-E1$_{159-166}$; TQFIFGPL), we detected SFV-specific CD8$^+$ T cell responses in the brain by stimulating CD8$^+$ T cells ex vivo and assessing IFNγ production (Figs. 2c and S1e). Following the cognate peptide stimulation, IFNγ-secreting CD8$^+$ T cells were found in the brains of SFV mono-infected mice, but not in the brains of mice infected with IAV only. Interestingly, in the SFV→IAV group, IFNγ was produced in response to stimulation with either IAV- or SFV-specific peptides, confirming the presence of IAV-specific T cells in the brain of mice previously exposed to neurotropic SFV.

To investigate the activation profile of CD8$^+$ T cells, we examined key T cell activation markers (CD25, CD38, KLRG1, PD-1) on total CD8$^+$ T cells and IAV tetramer$^+$CD8$^+$ T cells from lungs, brain and spleen of IAV-infected and co-infected mice (Fig. 2d). In the brain, we found significantly higher frequencies of CD8$^+$ T cells expressing effector KLRG1$^+$ or PD-1$^+$ molecules in the IAV only group (Fig. 2e). Conversely, SFV→IAV animals showed higher frequencies of brain CD8$^+$ T cells displaying activated CD38$^+$, CD38$^+$CD25$^+$ and CD38$^+$PD-1$^+$ phenotypes. In line with these findings, analysis of IAV tetramer-specific CD8$^+$ T cells also demonstrated that both D$^b$NP$_{366}$$^+$CD8$^+$ and D$^b$PA$_{224}$$^+$CD8$^+$ cells in the IAV-only group had significantly higher expression of the typical effector CD25 phenotype, whereas the SFV→IAV group had higher frequencies of CD38$^+$ D$^b$NP$_{366}$$^+$CD8$^+$ and CD38$^+$PD-1$^+$ D$^b$PA$_{224}$$^+$CD8$^+$ T cells. Thus, it is evident that activation profiles of brain IAV-specific CD8$^+$ T cells are quite distinct in mice with prior SFV infection (predominantly driven by CD38 expression) compared to a typical effector phenotype of CD8$^+$ T cells depicted by KLRG1, CD25 and PD-1 in IAV-only infected mice.

In the lung, the activation phenotype of CD8$^+$ T cells was comparable between SFV→IAV and IAV-only groups, with the exception of increased expression of the KLRG1$^+$PD-1$^+$ phenotype in D$^b$PA$_{224}$$^+$CD8$^+$ T cells in the IAV-only group (Fig. 2e). There were no differences in the activation phenotypes of CD8$^+$ T cells found in spleen of IAV-only and sequentially infected mice.

**Increased clonotypic diversity of IAV-specific CD8$^+$ T cells during SFV→IAV sequential infection**

The diversity and composition of T cell receptor (TCR)-αβ clonotypes within virus-specific CD8$^+$ T cell populations substantially impacts functionality, immunodominance and protective efficacy of CD8$^+$ T cells[19,20]. To understand TCRαβ signatures of IAV-specific CD8$^+$ T cells trafficking to the brain, relative to typical TCRαβ clonotypes of IAV-specific CD8$^+$ T cells trafficking to the infected lungs[21–23], we dissected the TCRαβ repertoire diversity and clonal composition using ex vivo tetramer staining and single-cell TCRαβ multiplex RT-PCR[24]. To the best of our knowledge, TCRαβ analysis of IAV-specific CD8$^+$ T cells in the brain has not been previously reported. We analysed TCRαβ repertoires of IAV-specific D$^b$NP$_{366}$$^+$CD8$^+$ and D$^b$PA$_{224}$$^+$CD8$^+$ T cells obtained from lungs and brains of SFV→IAV or IAV-only infected mice on d30 after IAV infection.

It is well-established that in the lungs of IAV-infected mice, both D$^b$NP$_{366}$$^+$CD8$^+$ and D$^b$PA$_{224}$$^+$CD8$^+$ T cell populations have strong biases towards specific TRBV gene segments, and to a lesser extent, towards TRAV gene elements. D$^b$PA$_{224}$$^+$CD8$^+$ TCRs in IAV-infected B6 mice display a bias towards TRAV6 and TRBV29 gene segments, whereas D$^b$NP$_{366}$$^+$CD8$^+$ TCRs predominantly utilise TRAV16 and TRBV13[23,25]. TCRβ repertoire within D$^b$NP$_{366}$$^+$CD8$^+$ T cells in IAV-infected mice is restricted and utilises public (shared across multiple mice) TCRβ signatures, especially TRBV13.1-CDR3β regions with prominent S/R/A/K GGA, GGS or GGG motifs[21]. In contrast, TCRβ repertoire within D$^b$PA$_{224}$$^+$CD8$^+$ T cell responses is diverse and private in nature (TCR clonotypes are not shared across the mice), although still use strong TRBV29 motifs[23,25].

In our study, we validated the dominant TRBV and TRAV usage for both D$^b$NP$_{366}$$^+$CD8$^+$ and D$^b$PA$_{224}$$^+$CD8$^+$ T cell populations in lungs of IAV-infected mice (Fig. 3a, b and Supplementary Tables 1–3). Using the TCRdist pipeline[26] for modelling of amino acid motifs (Figs. 3a, S2a, S3a and Supplementary Tables 1 and 2), we found strong TRBV13.1 bias with prominent public CDR3β motifs S/R/A/K GGA, GGS or GGG within D$^b$NP$_{366}$$^+$CD8$^+$ repertoires of IAV-infected mice, not only at the site of infection, but also in the brain. This provides evidence that the optimal public D$^b$NP$_{366}$$^+$CD8$^+$ TCRs can traffic to brain and establish memory pools at d30 following IAV infection (Fig. 3a). However, strikingly, it was apparent that D$^b$NP$_{366}$$^+$CD8$^+$ TCRαβ repertoire was more diverse in lungs and the brain of SFV→IAV sequentially infected mice, with the additional TRBV2-CDR3β DRRNS motif having a strong presence (Figs. 3a and 3c). TCRβ clonotypes with the TRBV2-CDR3β DRRNS motif generally play a very minor role in D$^b$NP$_{366}$$^+$CD8$^+$ TCR repertoire in IAV-infected B6 mice, largely dominated by TRBV13.1 public clonotypes. Interestingly, however, we have previously detected a strong presence of D$^b$NP$_{366}$$^+$CD8$^+$ clonotypes with the TRBV2-CDR3β DRRNS motif in A7 mice characterised by its fixed OT-I TCR-alpha chain[27]. In that study, we also found that the D$^b$NP$_{366}$$^+$CD8$^+$ T cell responses with the dominant TRBV2-CDR3β DRRNS motif were suboptimal in terms of their function, as depicted by peptide-MHC-I avidity and cytokine production[27]. Our findings thus suggest that markedly elevated cytokine milieu and/or prolonged IAV viral load in SFV→IAV sequentially infected mice led to the recruitment of suboptimal D$^b$NP$_{366}$$^+$CD8$^+$ TCR clonotypes to both lungs and brain, alongside public optimal D$^b$NP$_{366}$$^+$CD8$^+$ TCR signatures. Interestingly, in SFV→IAV mice, we also observed an increased usage of clonotypes using TRBV17 gene for D$^b$NP$_{366}$$^+$CD8$^+$ T cells (Fig. 3c), which was previously reported as a dominant TRBV in the naïve D$^b$NP$_{366}$$^+$CD8$^+$ TCR repertoire, generally not recruited into the IAV-specific immune response[23]. Thus, our findings demonstrating recruitment of suboptimal and 'naïve' D$^b$NP$_{366}$$^+$CD8$^+$ TCR clonotypes following SFV→IAV sequential infection might, at least in part, explain delayed viral control in SFV→IAV mice capable of recruiting TCR clonotypes of suboptimal functionality.

Similar findings were observed for diverse and private D$^b$PA$_{224}$$^+$CD8$^+$ TCR repertoire. For D$^b$PA$_{224}$$^+$CD8$^+$ TCR clonotypes in the lung, TRBV29 was the dominant gene segment within TCRβ for both

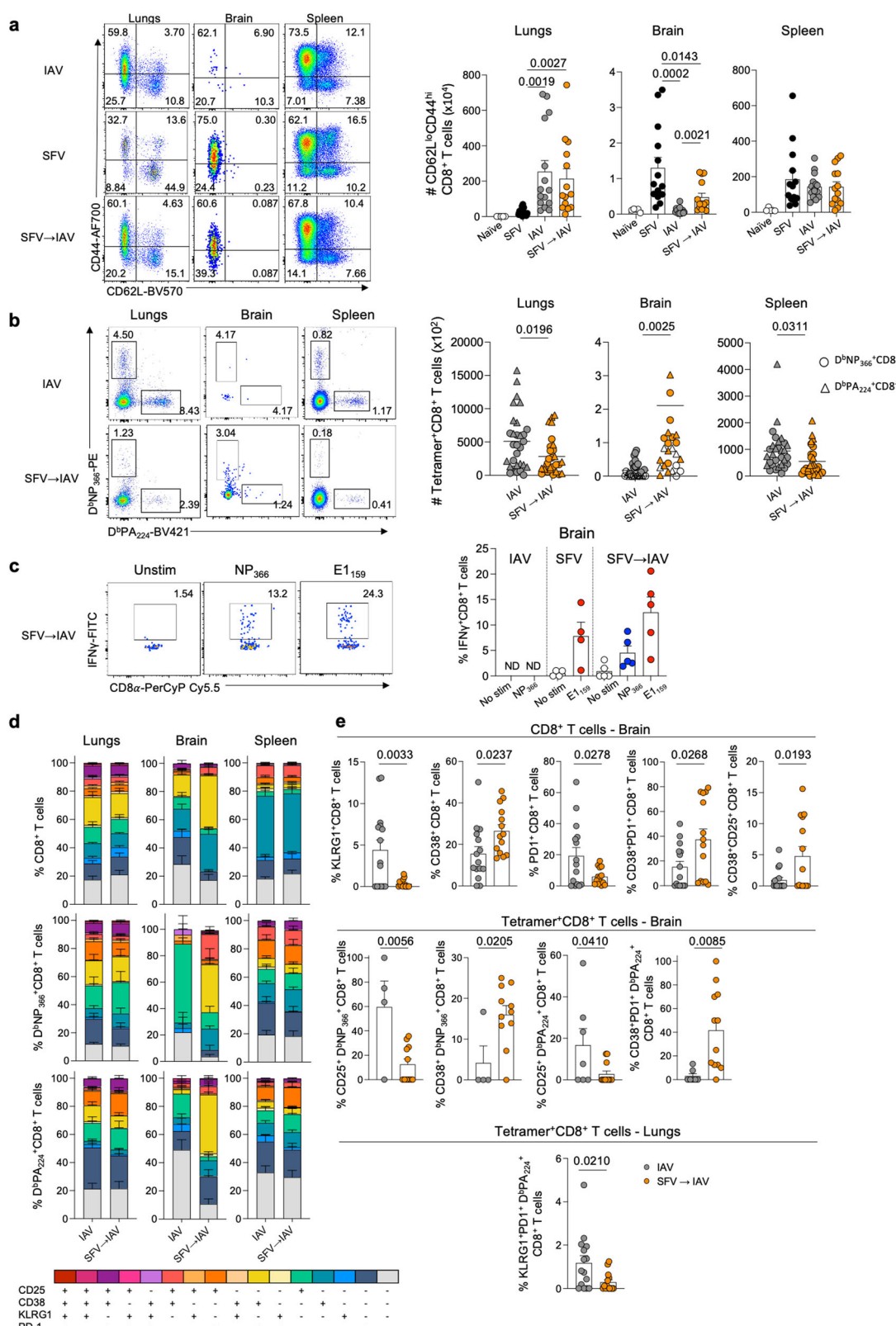

IAV-only and SFV→IAV sequentially infected mice (Fig. 3b). The SFV→IAV mice had, however, increased diversity within TRBV segments (Fig. 3c) and within the TRBV29-CDR3β motif (Fig. 3b). In the lung, D[b]PA$_{224}$[+]CD8[+] TCRs in the IAV group showed a bias towards TRAV6 and TRAV12, the latter also shown to be a dominant TCRα chain[28]. However, in SFV→IAV mice, D[b]PA$_{224}$[+]CD8[+] TCRs clearly demonstrated a more diverse TRBV pairing with a broad range of TRAV

segments (Fig. 3c and Supplementary Tables 1 and 3). TRAV6 and TRAV6/DV9 usage by D[b]PA$_{224}$[+]CD8[+] T cells was more pronounced at the expense of TRAV12 usage. Similar to what we observed for D[b]NP$_{366}$[+]CD8[+] TCRs, D[b]PA$_{224}$[+]CD8[+] repertoire was also more diverse in the brain compared to lungs. While there was still a bias towards TRBV29 for D[b]PA$_{224}$[+]CD8[+] TCRs in IAV-only group, TRBV usage was more heterogenous in the brain compared to that in lungs.

**Fig. 2 | SFV→IAV sequential infection perturbs influenza-specific CD8⁺ T cell responses. a** Absolute numbers of activated (CD44⁺CD62Lˡᵒ) CD8⁺ T cells across different anatomical sites in naïve ($n = 5$), SFV ($n = 14$), IAV ($n = 15$) and SFV→IAV ($n = 14$) infected mice. Representative FACS plots shown for each group and tissue; error bar represents SEM. Gating strategy is shown in Supplementary Fig. S5c. **b** Absolute numbers of IAV-specific CD8⁺ T cells directed at DᵇNP₃₆₆ and DᵇPA₂₂₄ epitopes across different anatomical sites in IAV ($n = 15$) and SFV→IAV ($n = 14$) infected mice. Open symbols represent data points where less than 10 cells were analysed. Representative FACS plots shown for each group and tissue; error bar represents SEM. **c** Intracellular cytokine staining of CD8⁺ T cells isolated on 7 dpi from the brain of IAV ($n = 5$), SFV ($n = 4$) or SFV→IAV ($n = 5$) infected mice, and stimulated with a cognate peptide. Mean frequencies of IFNγ producing cells are plotted. Representative FACS plots shown for SFV → IAV infection, error bars represent SD. **d** Stacked bar graphs depicting frequencies of activation marker combinations on CD8⁺ T cells (upper panel), DᵇNP₃₆₆⁺CD8⁺ T cells (middle panel) and DᵇPA₂₂₄⁺CD8⁺ T cells (bottom panel) on 7 dpi in IAV ($n = 15$) and SFV→IAV ($n = 14$) infected mice; error bar represents SEM. **e** Comparison of significantly different activation marker combinations in the brain and lungs (IAV $n = 15$, SFV→IAV $n = 14$, error bar represents SEM). Significance was determined by unpaired two-tailed Student's $t$ test. Source data are provided as a Source Data file.

We also used Simpson's Diversity Index (SDI) to define diversity of CDR3α and CDR3β chain sequences in the brain and lungs for DᵇNP₃₆₆⁺CD8⁺ and DᵇPA₂₂₄⁺CD8⁺ TCRs (Fig. 3d). The SDI results verified significantly greater diversity of DᵇPA₂₂₄⁺CD8⁺ TRAV and TRBV segments in the brain of SFV→IAV sequentially infected group as well as IAV-only group (CDR3β only). In addition, the DᵇPA₂₂₄⁺CD8⁺ TRBV compartment in the brain showed greater diversity in SFV → IAV than IAV-only.

Overall, our data demonstrated that while a substantial proportion of DᵇNP₃₆₆⁺CD8⁺ and DᵇPA₂₂₄⁺CD8⁺ TCRαβ repertoires within SFV→IAV sequentially infected mice have prototypical TCRαβ signatures of IAV-only infected mice, additional TCRαβ clonotypes are recruited after SFV→IAV infection, especially in the brain, with more extensive pairing of TRAV and TRBV. This in turn indicates a more heterogenous IAV-specific CD8⁺ T cell response following prior SFV infection, which however encompassed suboptimal TCRαβ clonotypes with diminished functions.

## IAV-specific T cell trafficking to the brain

To investigate the mechanisms underpinning increased trafficking of IAV-specific CD8⁺ T cells to the brain in SFV→IAV mice, we first assessed the integrity of blood-brain barrier (BBB) using Evans Blue Dye extravasation method[29,30]. The BBB is often compromised during infection and inflammation of the CNS[31]. Inflammatory cytokines associated with viral infections can increase permeability of the BBB[32]. Extraneural SFV infection is well-documented to disrupt the BBB[33]. To investigate the integrity of the BBB in our dual infection system, Evans Blue Dye was i.p. injected at predefined time-points post-infection. After 3 hrs, mice were culled, the brain vasculature perfused and brain dye levels determined. In agreement with a previous study, BBB permeability was significantly increased by SFV (i.p.) infection (Fig. 4a). Surprisingly, IAV infection also affected BBB permeability. While the changes were greatest for each infection at d5, both infections also showed increased permeability as early as d3 and remained permeable at d7. Mice were then infected with IAV (i.n.) on d7 post-SFV infection. Interestingly, a day later we found a significant increase in BBB permeability, which continued the second day, although at reduced magnitude. These SFV-induced BBB changes could be facilitating IAV and CD8⁺ T cell entry into the brain.

To differentiate whether IAV-specific CD8⁺ T cells trafficked to the brain because (i) IAV antigen present in the brain (antigen-specific recruitment) or (ii) as a consequence of the ongoing SFV neurotropic infection (non-antigen-related recruitment), we utilised a well-established transgenic OT-I system and infection with IAV-OVA[34,35]. Enriched OT-I CD8⁺ T cells were activated in vitro by co-culturing for 4 days with SIINFEKL peptide-pulsed naïve splenocytes from a B6 mouse[36]. 1x10⁶ of in vitro activated OT-I cells were transferred into naïve mice or mice previously infected with SFV on d7 post-infection. One day after the transfer, the recipient mice were infected with 1x10⁴ pfu of either IAV-OVA or wild-type IAV. Organs were harvested on d7 after the IAV infection (Fig. 4b). CD8⁺ OT-I cells were detected in the brain and lungs of mice infected with the IAV-OVA (either single infection or sequential infection), but not in mice infected with IAV (wild-type X31; Fig. 4c), indicating that OT-I cells could traffic to the site where OVA antigen was likely present. We performed the same experiment using naïve (rather than in vitro activated) OT-I cells and demonstrated that OT-I cells also traffic to the brain of IAV-OVA and SFV→IAV-OVA infected animals but were absent in the IAV control group (Fig. S1f).

Our findings indicate that OT-I cells specifically trafficked to the brain when their cognate OVA antigen was present in the mice. To determine whether IAV was present in the brain, we performed qPCR amplification of the IAV matrix (M) gene region in perfused brains. Our data indicate that IAV RNA was present in the brain of IAV and SFV→IAV infected mice during acute phase of infection (Fig. 4d). Expression of the virus M gene peaked on d5 and levels were higher in the brains of SFV→IAV mice than in those of IAV only mice. Interestingly, RNA expression kinetics mirrored the difference in viral kinetics observed in the lungs (Fig. 1b). Despite readily detectable IAV RNA level in the brain, we could not detect infectious virus by standard plaque assay, presumably due to low sensitivity of this method.

It seems most likely that IAV and OVA-specific T-cells traffic to the brain as a result of the presence of these antigens in the brain. Our data indicate that prior SFV infection may have long-term effect on the integrity of BBB, and it also modulates subsequent brain and lung immune responses.

## Surface proteome analysis of IAV-specific CD8⁺ T cells in lungs and brain

To define in-depth characteristics of CD8⁺ T cells trafficking to different anatomical sites in IAV and IAV→SFV mice, we performed an extensive analysis of 253 cell surface antigens on OT-I cells following IAV-OVA infection with or without prior SFV infection. We utilised LEGENDscreen™ and OT-I system to perform immunoprofiling of CD8⁺ T cells. OT-I cells isolated from brain, lungs, and spleen of IAV-OVA-only infected and sequentially infected SFV→IAV-OVA mice were barcoded with anti-mouse CD45.1 antibodies, stained with CD8⁺ T cell backbone panel, and subsequently screened for the expression of 253 cell surface PE-labelled markers using flow cytometry (Fig. 5a). We then used the InfinityFlow pipeline to predict co-expression of individual PE-labelled markers[37]. Data from each well were combined with the information contained in the backbone panel to enable data imputation into a data matrix, in order to predict marker expression by CD8⁺ T cells.

We performed UMAP analysis of lung CD8⁺ T cells using markers from the backbone panel and identified five main clusters corresponding to 1) terminally differentiated effectors, 2) T_RM/T_EX-like, 3) T_EFF1, 4) T_CM-like and 5) T_EFF2 cells (Fig. 5b)[38]. No major differences were observed between cells isolated from IAV-OVA or SFV→IAV-OVA infected animals in terms of phenotypes. However, when CD8⁺ T cells from each group were visualised on separate UMAPs (Fig. 5c), we found reduced frequency of terminally differentiated effector OT-I cells (cluster 1) within the SFV→IAV mice, resulting in an approximate ratio of 70:30 (IAV-OVA:SFV→IAV-OVA) (Fig. 5d). This corroborated our earlier findings (Fig. 3b), where we observed a significantly lower numbers of IAV-specific CD8⁺ T cells in the SFV→IAV group. To confirm the identification of OT-I cells in cluster 1, we scrutinised a number of

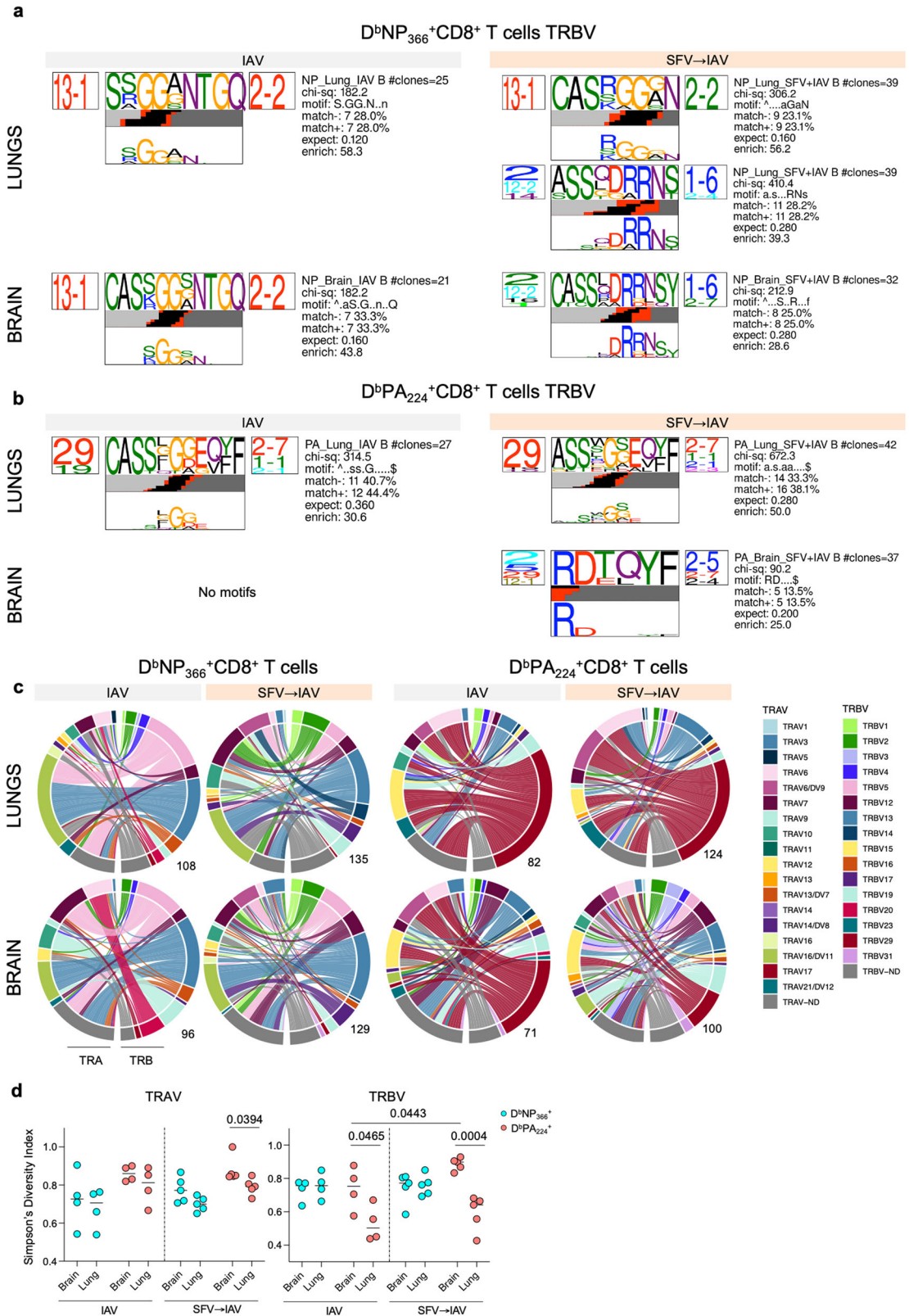

**a** $D^bNP_{366}^+CD8^+$ T cells TRBV

**b** $D^bPA_{224}^+CD8^+$ T cells TRBV

**c** $D^bNP_{366}^+CD8^+$ T cells          $D^bPA_{224}^+CD8^+$ T cells

**d**

cell surface markers associated with terminally differentiated effector T cells (KLRG1, CX3CR1, CD43, integrin B7, CD11b, and CD11c) and confirmed their markedly higher expression on OT-I cells in terminally differentiated cluster 1, compared to other clusters (Fig. 5e). Despite collating CD8+ T cells from 16 brains, we did not obtain sufficient CD8+ T cell numbers to allow meaningful comparisons between the two groups.

**Prior SFV infection induces paralysis of antigen presenting cells and consequently suboptimal proliferation of CD8+ T cells**

Our data provide evidence that SFV→IAV sequentially infected mice have delayed IAV lung clearance (Fig. 1b), and reduced numbers of IAV-specific tetramer+CD8+ T cells in the lungs (Fig. 2b). To understand the underlying mechanisms governing differential activation and trafficking of IAV-specific CD8+ T cells, linked at least in part to delayed viral

**Fig. 3 | Increased TCRαβ repertoire clonotypic diversity of IAV-specific CD8⁺ T cells during SFV→IAV sequential infection.** TCR representation of CDR3β sequence motifs within (**a**) DᵇNP₃₆₆⁺CD8⁺ T cells and (**b**) DᵇPA₂₂₄⁺CD8⁺ T cells in the lungs and brains of IAV and SFV → IAV mice. Each TCR chain motif depicts the V (left side) and J (right side) gene frequencies, the CDR3 amino acid sequence (middle), and the inferred rearrangement structure (bottom bars coloured by source region; V-region, light grey; insertions, red; diversity (D)-region, black; and J-region dark grey). The motif scores were determined by chi-squared, with values greater than 90 considered highly significant. **c** Circos plots of TRAV and TRBV clonotype pairings for DᵇNP₃₆₆ and DᵇPA₂₂₄ epitopes in the lung and brain of IAV and SFV → IAV infected mice. Left arch and segment colours show TRAV usage, while the outer right arch depicts TRBV usage. Each coloured segment indicates TCRαβ clonotypes with the same V gene segment usage, but not CDR3 sequences. The number on the bottom right shows the number of sequences considered for each circos plot. **d** Simpson's diversity index (SDI) analysis of TRAV and TRBV usage in the lungs and brains of IAV ($n = 4$) and SFV → IAV ($n = 5$) in DᵇNP₃₆₆ and DᵇPA₂₂₄ epitopes CD8⁺ T cells. Significance was determined by unpaired two-tailed Student's t-test. Source data are provided as a Source Data file.

clearance, we sought to define proliferative capacity of IAV-specific CD8⁺ T cells. Impaired T cell activation and trafficking has been previously linked to the phenomenon of antigen-presenting cell (APC) paralysis[39], resulting from an excessive production of immunomediators directed at restoration of homeostasis. Our data (Fig. 1c) showed markedly elevated levels of proinflammatory cytokines and chemokines in the lungs of sequentially infected SFV→IAV animals. SFV could be detected in the lungs on day 3 post-infection, and we also detected the presence of SFV-specific CD8⁺ T cells in the lungs of mice exposed to SFV or SFV→IAV (Fig. 6a). Thus, we hypothesised that SFV causes APC paralysis, which could subsequently impact CD8⁺ T cells activation and proliferating capacity. To investigate the possibility of APC paralysis, we inoculated mice with either SFV, IAV or, as a control, with a Toll-like receptor agonist, CpG. The latter induces APC paralysis in the spleen[39]. Cell surface markers associated with APC paralysis (Ly6A, CD103, TGFβRII and PD-L1[40]) were analysed on splenic DCs (Fig. 6b). Our data showed that SFV infection-induced upregulation of surface expression of Ly6A on splenic CD8⁺ DCs, a phenotype consistent with paralysis, akin to that induced by CpG (Fig. 6c).

As impairment in APC antigen-presenting function can impact T cell activation and proliferation[41], we investigated the impact of prior SFV infection on the proliferative capacity of IAV-specific T cells using the OT-I system. SFV-infected or naïve mice received 1x10⁶ VPD450-labelled OT-I cells, and subsequently were infected with IAV-OVA at d1 after OT-I transfer (Fig. 6d). The proliferative capacity of OT-I cells was assessed on day 3.5 in the draining mediastinal lymph node (MLN), and numbers of OT-I cells in the lungs were assessed on day 4.5 post-IAV-OVA infection. Prior SFV infection impaired division of OT-I CD8⁺ T cells in the MLN relative to IAV-OVA infection alone (Fig. 6e). OT-I CD8⁺ T cells from the SFV→IAV-OVA group proliferated significantly slower than those from the IAV-OVA mice. The proportion of proliferating cells in early divisions (3-5) were significantly higher in the SFV→IAV-OVA group, while fewer OT-I cells underwent the 7+ divisions. This is consistent with suboptimal APC activation impacting their proliferative capacity (Fig. 6e). As a result, significantly reduced numbers of activated OT-I cells were present in the infected lungs in SFV→IAV-OVA mice at day 4.5 after IAV infection (Fig. 6f).

Collectively, our findings provide clear evidence that prior infection with SFV can negatively affect the APC function and antigen presentation efficacy. This impairment in DC function subsequently led to suboptimal IAV-specific CD8⁺ T cell activation/division, reduced IAV-specific CD8⁺ T cell numbers trafficking to the infected lung, thus diminished protective capacity of IAV-specific CD8 + T cell responses following sequential SFV→IAV sequential-infection.

### Sequential SFV→IAV infection impacts IAV-specific CD8⁺ T cell memory formation and recall capacity

Given the impact of prior SFV infection on the activation and magnitude of a subsequent IAV CD8 T-cell response and on the resolution and pathology of the acute lung infection, we hypothesised that prior SFV infection also impairs establishment and long-term persistence of IAV-specific CD8⁺ memory T cells. In addition to IAV and SFV→IAV groups, we included IAV→SFV group in our experiments to determine whether the order of sequential infections has an effect on memory CD8⁺ T cells. Mice were initially i.n. infected with 10⁴ pfu IAV (x31) or

i.p. 5×10³ pfu SFV (A7(74)), and then infected with a second virus 8 days later (Fig. 7a). Remarkably, ninety days after the second infection, we detected sizeable populations of IAV-specific memory CD8⁺ T cells of both specificities, DᵇNP₃₆₆⁺CD8⁺ and DᵇPA₂₂₄⁺CD8⁺ T cells, in the brains of all experimental groups (Fig. 7b). Tetramer⁺ CD8⁺ T cell numbers were significantly elevated in SFV→IAV group compared to mice infected with IAV only, but not in the IAV→SFV group (Fig. 7b). More specifically, we observed significantly lower numbers of T_EM and T_CM cells in the spleen of IAV→SFV sequentially infected mice, but higher numbers of T_EM in the lungs of SFV→IAV group (Figs. 7c–e and S4a, b). Overall, memory tetramer⁺CD8⁺ T cell numbers in the lungs were comparable between IAV-only and SFV→IAV mice, which is in a stark contrast to the acute phase of influenza infection. There was also a significant reduction in tissue resident memory (T_RM) CD69⁺CD103⁺CD8⁺ T cells in the lungs of the IAV→SFV co-infected group (Fig. 7f), and an apparent presence of IAV-specific CD69⁺CD103⁻ and CD69⁺CD103⁺ T_RM cells[42] in the brain across all the experimental groups, regardless of modality of infection. SFV→IAV mice had significantly higher numbers of T_RM cells compared to IAV-only group (Fig. 7f).

To determine whether sequential SFV→IAV infection also affects the recall capacity of IAV tetramer⁺CD8⁺ T cell responses, we used the prime and challenge system, which utilises two serologically different IAV strains. Mice were initially infected with either 10⁴ pfu x31 H3N2 IAV (i.n.) or 5 × 10³ pfu A7(74) SFV (i.p.), followed on d8 by infection with either i.p. SFV (IAV→SFV) or i.n. IAV (SFV→IAV), respectively (Fig. 8a). IAV-only group was used as a control. All primed mice were subjected to a subsequent i.n. infection with 10³ pfu H1N1 PR8 at 100 days post-co-infection. At d5 following IAV-PR8 challenge, lung, brain and spleen tissues were harvested from all experimental groups. Viral loads in lungs were determined by plaque assay and cytokine levels measured by Legendplex. No differences were detected in either weight loss, lung virus titres or cytokine/chemokine milieu between the experimental groups (Fig. S4d–f), indicating that all groups could mount optimal recall IAV-specific tetramer⁺CD8⁺ T cell responses against the secondary PR8 IAV infection. However, a significant reduction of IAV-specific tetramer⁺CD8⁺ T cells was found in lungs of the [IAV→SFV]→PR8 group (Fig. 8b), relative to the [IAV]→PR8 control, while the [SFV→IAV]→PR8 group had comparable responses to the [IAV]→PR8 group. Remarkably, there was a significantly higher number of tetramer⁺CD8⁺ T cells in the brains of the [IAV]→PR8 group, compared to the [SFV→IAV]→PR8 or [IAV→SFV]→PR8 groups. Moreover, the magnitude of tetramer⁺CD8⁺ T cell recall response in the lungs was significantly lower in [IAV→SFV]→PR8 (Fig. 8c). Despite these differences, animals across all experimental groups displayed the same level of virus control in the lungs. Interestingly, brain CD8⁺ T cell recall responses were also significantly lower in the [SFV→IAV]→PR8 and [IAV→SFV]→PR8 groups when compared to the [IAV]→PR8 group (Fig. 8c).

Following the primary IAV infection in C57BL/6 mice, DᵇNP₃₆₆⁺CD8⁺ and DᵇPA₂₂₄⁺CD8⁺ T cell responses are co-dominant, while DᵇNP₃₆₆⁺CD8⁺ T cell responses numerically dominate following the secondary IAV infection[18,43,44]. This well-described immunodominance hierarchy following secondary IAV infection was apparent in [IAV]→PR8 and [SFV→IAV]→PR8 experimental groups in both lungs and

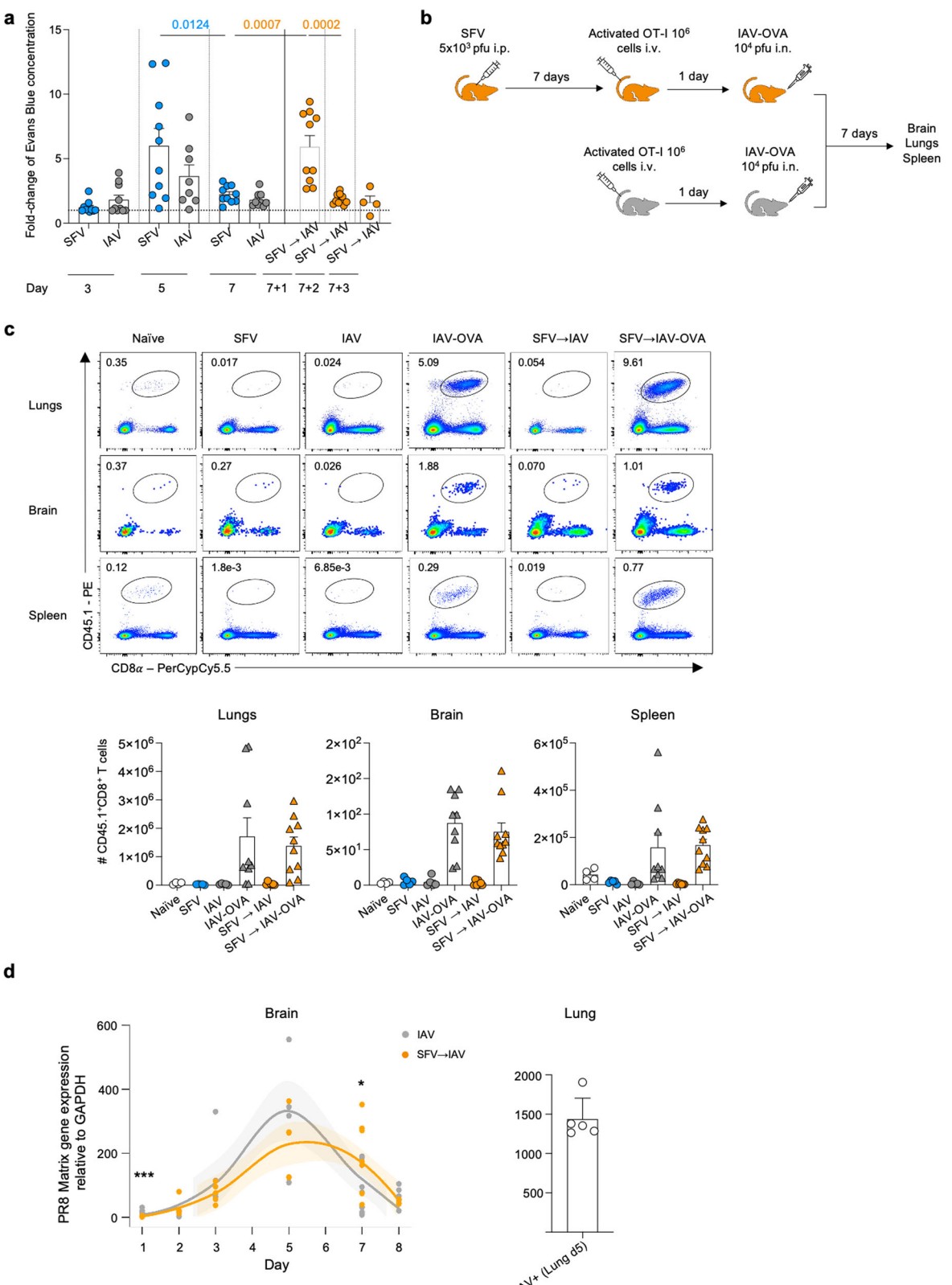

brain (Fig. 8d). However, the immunodominance of $D^bNP_{366}^+CD8^+$ T cell responses was completely abolished in [IAV→SFV]→PR8 group, where lung and brain $D^bNP_{366}^+CD8^+$ T cell responses were numerically equivalent to $D^bPA_{224}^+CD8^+$ T cells. The overall frequencies of effector $CD44^{hi}CD62L^{lo}$ tetramer$^+$ CD8$^+$ T cells were comparable in lungs, spleens and brain (Fig. 8e and S4c). However, the absolute numbers of activated $CD44^{hi}CD62L^{lo}$ tetramer$^+$ CD8$^+$ T cells were significantly

reduced in the lungs of [IAV→SFV]→PR8 group and in the brain of [SFV→IAV]→PR8 and [IAV→SFV]→PR8 groups (Fig. 8e).

Overall, our findings demonstrate that SFV infection has long-term effects on IAV-specific CD8$^+$ T cell memory generation and recall responses. This was particularly apparent in IAV→SFV, which affected both long-term memory and recall responses in the lungs. Yet, despite the overall reduction in magnitude and the abolition of

**Fig. 4 | Factors driving IAV-specific T cell trafficking to the brain. a** Evans blue (EB) extravasation assay in mice infected with either SFV or IAV or SFV→IAV co-infected, compared to naïve controls. EB was injected i.p. on different days, brains were perfused and harvested 3 h post-injection. EB concentration was measured spectrophotometrically in brain homogenates and determined using a standard curve. Data from 2 independent experiments were normalised to naïve controls and plotted, error bar represents SEM (d3: SFV $n = 10$, IAV $n = 10$; d5: SFV $n = 10$, IAV $n = 8$; d7: SFV $n = 10$, IAV $n = 9$; d7 + 1: SFV → IAV $n = 10$; d7 + 2: SFV → IAV $n = 10$; d7 + 3: SFV → IAV $n = 4$). **b** Mice were infected i.p. with $5 \times 10^3$ pfu A7(74) SFV. $1 \times 10^6$ naïve or in vitro activated CD8$^+$ OT-I T cells were adoptively transferred i.v. to 7 days after infection SFV mice and uninfected mice. D1 post transfer all mice were infected i.n. with $1 \times 10^4$ pfu A/x31-OVA. Brain, lungs, and spleen were collected at 7 days after infection. IAV, SFV, and SFV → IAV has been included as control (not shown) and has also received naïve or in-vitro activated OT-I T cells. **c** Absolute

numbers of OT-I T cells across different anatomical sites in naïve ($n = 4$), SFV ($n = 5$), IAV ($n = 5$), SFV → IAV ($n = 7$), IAV-OVA ($n = 9$) and SFV → IAV-OVA ($n = 10$) with in vitro activated OT-I T cell transfer (error bar represents SEM). Representative FACS plots shown for each group and tissue. Gating strategy is shown in Supplementary Fig. S5d. **d** A/PR8 (H1N1) matrix (M) gene expression in the brain of IAV-only and SFV → IAV infected mice across different timepoints. Gene expression is normalised to GAPDH and the ΔΔCt method used to calculate relative gene expression. Mean data are shown for biological replicates (d1: IAV $n = 5$, SFV → IAV $n = 5$; d2: IAV $n = 5$, SFV → IAV $n = 5$; d3: IAV $n = 5$, SFV → IAV $n = 5$; d5: IAV $n = 4$, SFV → IAV $n = 5$; d7: IAV $n = 9$, SFV → IAV $n = 8$; d8: IAV $n = 5$, SFV → IAV $n = 5$; error bar represents SEM). Shaded regions indicate 95% confidence interval. A/PR8 M gene expression in the lungs of IAV infected mice ($n = 5$) at 5 dpi was included as positive control. P values d1 *** = 0.0008, d7 * = 0.0464. Significance was determined by unpaired two-tailed Student's t-test. Source data are provided as a Source Data file.

immunodominance hierarchy, mice in the [IAV→SFV]→PR8 group did not show signs of exacerbated respiratory disease, at least up to d5 after infection. Moreover, our experiments show that the H1N1-x31 strain of IAV enters the brain, as evidenced by the presence of viral RNA and T$_{RM}$ CD8$^+$ T cells, promoted by the presence of antigen in the non-lymphoid tissues[45].

## Discussion

Humans are continuously exposed to multiple pathogens at any given time, and several studies suggest that co-infections with unrelated pathogens are a common event and can alter the progression of disease[46]. Yet, our understanding of consequences of co-infections remains rudimentary. It has long been recognised, that viral respiratory infections predispose patients to bacterial infections, and these co-infections have a worse outcome than either of the single infection. Such co-infections have been particularly well documented for influenza[47] and more recently SARS-CoV2[48]. On the other hand, there is a paucity of data on co-infections with unrelated viruses, which are rarely being studied[49,50]. To comprehensively understand the complex nature of immune responses during viral co-infection, we established a mouse model to study sequential infections with two unrelated viruses in two key tissues, the lung and the brain. Influenza viruses and encephalitic viruses are co-endemic globally[51] and although the frequency of such co-infections is likely to be high, data are scarce. The complexity of the issue is compounded by the fact that majority of encephalitic viral infections is asymptomatic[52]. Our study reveals compelling findings that delivery of SFV prior to IAV (SFV→IAV) has a detrimental effect on immune responses to influenza viruses. It markedly skews dynamics of immune responses to IAV, resulting in altered T cell trafficking, higher viral load in the lungs, aggravated lung histopathology and affects magnitude, TCR clonal distribution and inflammatory milieu during respiratory virus infection. While this study focused on responses to IAV, it is reasonable to anticipate perturbed immune responses to SFV following IAV infection. These investigations are currently ongoing.

Initial observations revealed a significantly lower viral load in lungs of SFV→IAV at the early stage of infection (3 dpi). One possible explanation for the early viral load suppression is the effect of type I IFN triggered by SFV infection. Type I IFN inhibits IAV replication[53] and SFV is a known inducer of early interferon response[54]. However, by day 7 post-infection these mice showed exacerbated respiratory disease characterised by dysregulated inflammatory responses and greater tissue damage in the lungs. Paradoxically, these have been linked to excessive IFNα/β signalling during acute influenza virus infection and influenza-bacterial co-infections[55,56]. While we found no changes in B cell and myeloid cell compartments in either lungs or brain, there was an unexpected influx of activated CD8$^+$ T cells to the brain of SFV→IAV mice. These cells were IAV-specific and appeared to traffic to the brain instead of lungs and spleen, both tissues showing significantly decreased immunodominant D$^b$NP$_{366}$$^+$ and D$^b$PA$_{224}$$^+$ T cell responses. A

significantly larger proportion of bulk CD8$^+$ T cells infiltrating the brain in SFV→IAV group was CD38$^+$PD-1$^+$, and these CD8$^+$ T cells were IAV tetramer-specific. On the other hand, the majority of CD38$^+$PD-1$^+$CD8$^+$ T cells infiltrating the brain of IAV-only infected group were IAV tetramer-negative, suggesting recruitment of bystander CD8$^+$ T cells to the brain in the absence of SFV infection, presumably due to their inflammation-induced expansion in the periphery. We previously reported that co-expression of CD38 and PD-1 on CD8$^+$ T cells characterises hyperactivation in severe influenza disease and is linked to higher viral load at the site of infection[57]. Although SFV→IAV infected animals demonstrated higher viral load, we did not observe any differences in the activation profile in the lungs. Interestingly, the majority of IAV tetramer-positive CD8$^+$ T cells infiltrating the brain in IAV-only group were CD25$^+$ and much higher proportion was D$^b$NP$_{366}$-specific rather than D$^b$PA$_{224}$-specific. This indicates recent CD8$^+$ T cell activation in draining lymph nodes, with the levels of CD25 expression on tetramer$^+$CD8$^+$ T cells being consistent with previous reports[58].

Our adoptive transfer experiments using naïve and activated OT-I cells indicated that trafficking of IAV-specific CD8$^+$ T cells to the brain is antigen-driven. Although we could not detect infectious influenza virus in the brain of either IAV or SFV→IAV infected animals, we detected IAV matrix RNA at high levels in SFV→IAV group. The viral entry into the brain and influx of IAV-specific CD8$^+$ T cells appears to be facilitated by increased BBB permeability. Both IAV and SFV could affect BBB permeability, but this was further enhanced by sequential SFV→IAV infection, as measured by Evans Blue dye extravasation. While SFV is known to compromise BBB[59], we used a non-neurotropic strain of influenza, which does not cause brain infection, although it has been shown to affect BBB permeability to some extent[60]. Therefore, the exacerbation of BBB leakiness has been compounded by IAV infection at the early phase post-co-infection.

The effects of SFV→IAV co-infection not only affected cell trafficking but had a multipronged effect on short- and long-term immunity. Primary infection (either bacterial or viral) can lead to the development of paralysed DCs and macrophages with poor antigen-presenting capabilities and diminished ability to secrete immunogenic cytokines[39], which in turn results in impaired T cell activation and trafficking. The immune paralysis of APCs is a result of an excessive production of local mediators directed at reduction of immuno-pathology caused by inflammatory responses. Our data show an excessive production of proinflammatory cytokines and chemokines in the lungs of SFV→IAV animals and significantly higher viral titres in the lungs. Consequently, we observed impaired proliferation of antigen-specific CD8$^+$ T cells in the MLN resulting in significantly reduced numbers of these cells in the lungs.

Our TCRαβ analysis revealed closely related TCRαβ clonotypes, with prominent IAV-specific TCR gene segments and motifs cluster according to their Vα, Vβ, Jα, and Jβ signatures, indicating selective recruitment of IAV epitope-specific T cells. These findings agree with studies that reported a biased TRAV and TRBV gene usage and

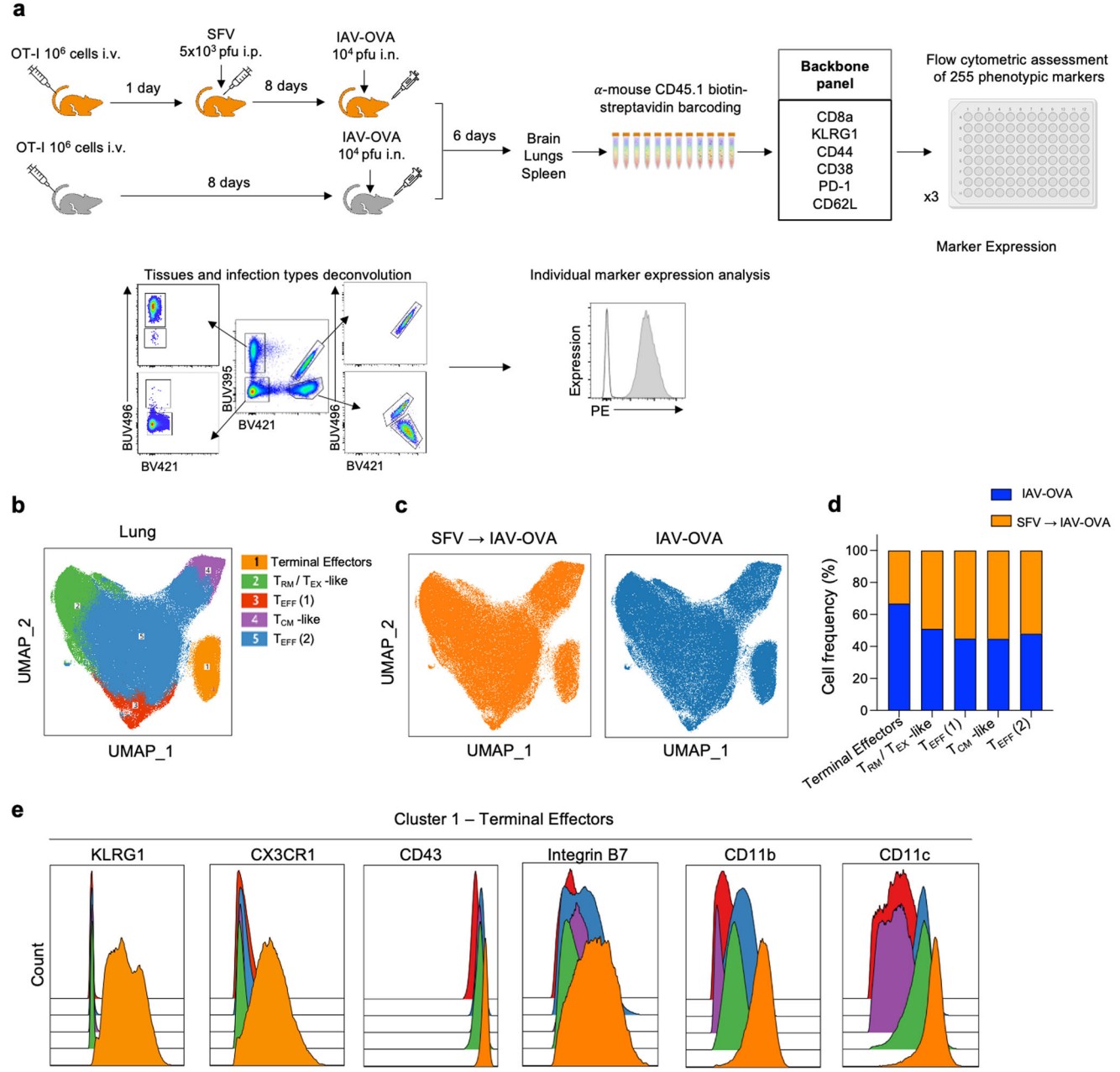

**Fig. 5 | Surface proteome analysis of CD8+ T cells migrating to the brain.**
**a** Schematic workflow for flow cytometric analysis of surface proteome.
CD45.1+CD8+ T cells were enriched from the brain, lungs, and spleen of IAV-OVA and
SFV → IAV-OVA infected mice, barcoded, stained with a CD8+ T cell focused back-
bone panel, then stained with phycoerythrin (PE) conjugated labelled antibodies
for surface proteome expression with the LEGENDScreen kit (Biolegend), followed
by individual expression analysis. **b** UMAP plot generated using InfinityFlow-
predicted expression profiles for indicated populations in lungs OT-I CD8+ T cells

from both SFV → IAV-OVA and IAV-OVA groups. **c** UMAP plot of OT-I CD8+ T cell
subsets in the lungs generated using backbone markers from SFV → IAV-OVA
(orange) and IAV-OVA (blue) groups. **d** Comparison of cell frequencies between
SFV → IAV-OVA and IAV-OVA groups in different clusters identified by InfinityFlow.
**e** Histogram representations of activation markers associated with CD8+ T cells:
KLRG1, CX3CR1, CD43, integrin B7, CD11b and CD11c in Cluster 1 (terminal effec-
tors). Source data are provided as a Source Data file.

prevalent motifs in IAV infection in B6 mice[18,21,23,28]. Prior to our study,
ex vivo epitope-specific TCR analysis in the brain during IAV infection
was non-existent. Our data showed a statistically significant increase in
clonotype diversity during SFV→IAV sequential infection, especially in
the brain, with a much greater and more extensive pairing of TRAV and
TRBV, indicating a more heterogenous CD8+ T cell response encom-
passing suboptimal TCRs[27]. Such increased diversity of TCRαβ clono-
types within IAV-specific CD8+ T cells raises the question about the
nature of TCRαβ clonotypes recruited into the immune response
exclusively during sequential infection with a neurotropic virus.

A surprising finding of our study is the abolition of the immuno-
dominance hierarchy in IAV→SFV group. While we observed significant
changes in CD8+ T cell responses to IAV during the acute phase in
SFV→IAV group, there was no difference in CD8+ T cell memory for-
mation or recall capacity. However, when the order of viruses was
reversed, we found a significantly reduced numbers of IAV
tetramer+CD8+ T cells in the lungs, and the magnitude of the recall
response was significantly diminished. Interestingly, the immunodo-
minance hierarchy was eliminated in IAV→SFV group. Nucleoprotein
$NP_{366}$ and acid polymerase $PA_{224}$ peptides presented by H2D$^b$ in

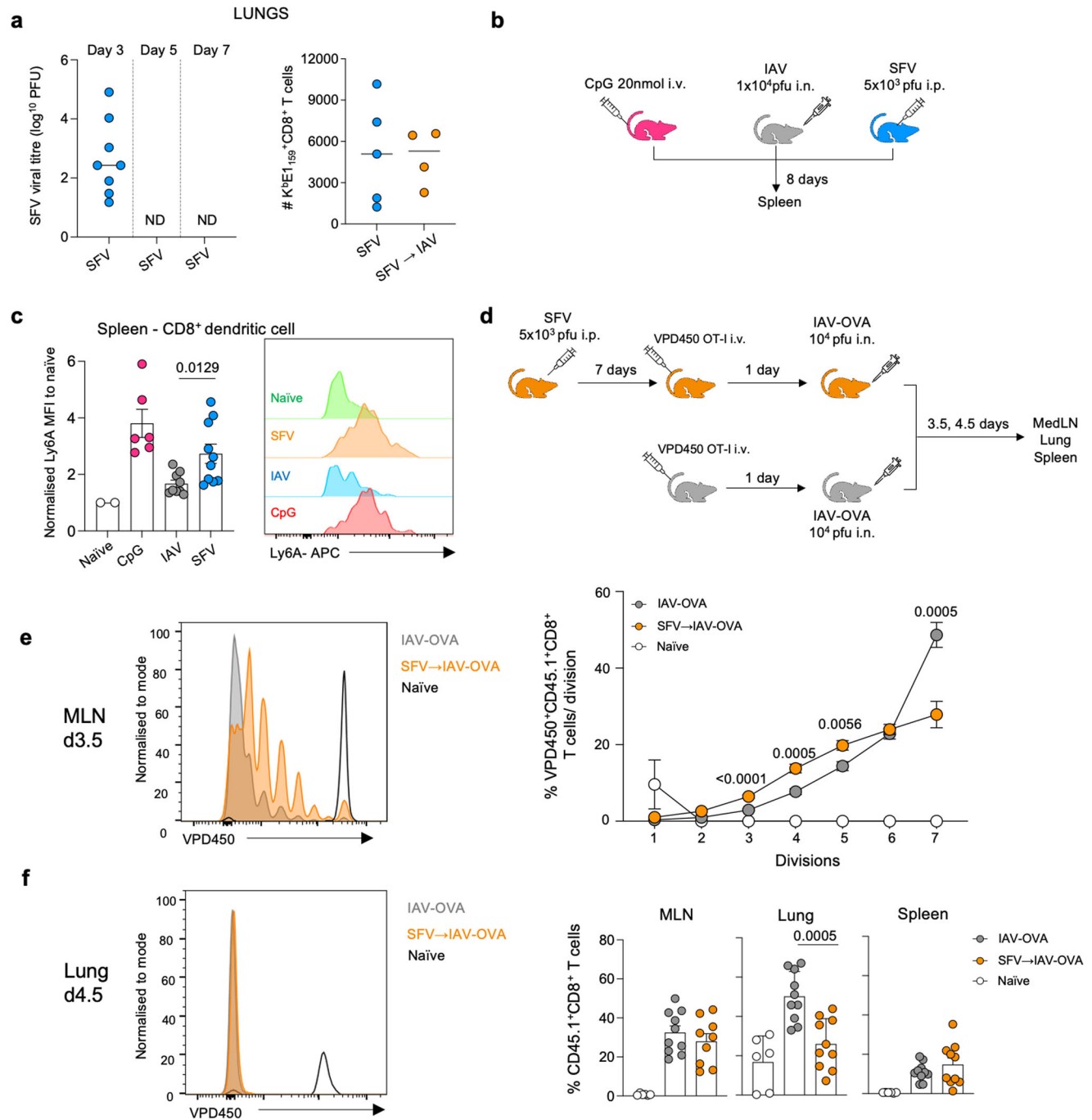

**Fig. 6 | SFV infection induces APC paralysis. a** Lung SFV viral titres were determined by a plaque assay on Vero cells on d3, d5 and d7 ($n = 8$). Absolute numbers of SFV-specific CD8$^+$ T cells of K$^b$E1$_{159}$ specificity in the lungs of SFV ($n = 5$) and SFV → IAV ($n = 4$) infected mice on d7. Each symbol denotes an individual mouse. **b** Mice were infected i.p. with 5×10$^3$ pfu A7(74) SFV, or with 10$^4$ pfu A/HKx31 (i.n.), or treated with 20-nmol CpG 1668 i.v., a TLR9 ligand. **c** Normalised mean fluorescence intensity of cell surface marker Ly6A (SCA–1) on CD8$^+$ dendritic cell population in spleen of mice infected with IAV ($n = 9$), SFV ($n = 10$), and CpG treated ($n = 6$) compared to naïve ($n = 2$). Representative histogram shown for expression of Ly6A on CD8$^+$ DCs in the spleen of each infection, error bar represents SEM. **d** Mice were infected i.p. with $5 \times 10^3$ pfu A7(74) SFV. $1 \times 10^6$ VPD450-labelled naïve CD8$^+$ OT-I T cells were adoptively transferred i.v. to SFV-infected mice on d7 and uninfected mice. D1 post transfer all mice were infected i.n. with $1 \times 10^4$ pfu A/x31-OVA. Mediastinal lymph node (MLN), lungs, and spleen were collected on d3.5 and d4.5. **e** Proportion of proliferating (VPD450 low) OT-I T cells at each division in the MLN on d3.5. Representative histogram shown for each division (naïve $n = 5$, IAV-OVA $n = 9$, SFV → IAV-OVA $n = 9$: error bar represents SEM). Gating strategy is shown in Supplementary Fig. S5d. **f** Proportion of divided (VPD450 low) OT-I T cells in the MLN, lungs, and spleen at 4.5 dpi. Representative histogram shown for OT-I T cells division (naïve $n = 6$, IAV-OVA $n = 10$, SFV → IAV-OVA $n = 10$; error bars represent SEM). Significance was determined by unpaired two-tailed Student's $t$ test. Source data are provided as a Source Data file.

C57BL/6 mice are characterised by well-defined immunodominance hierarchy. The primary CD8$^+$ T cell responses to D$^b$NP$_{366}$ and D$^b$PA$_{224}$ are of comparable magnitude, however, D$^b$NP$_{366}$$^+$CD8$^+$ T cell responses are by far immunodominant and constitute majority (80–90%) of influenza-specific CD8$^+$ T cells after the secondary IAV infection[61]. The main factor driving the immunodominance in the recall response is antigen presentation during the primary response[62] and antigen persistence of the NP$_{366}$ epitope[63]. CD8$^+$ T cells of different specificities receive discrete signals during the late phase of acute infection (post day 8) impacting the immunodominance hierarchy, and T cells

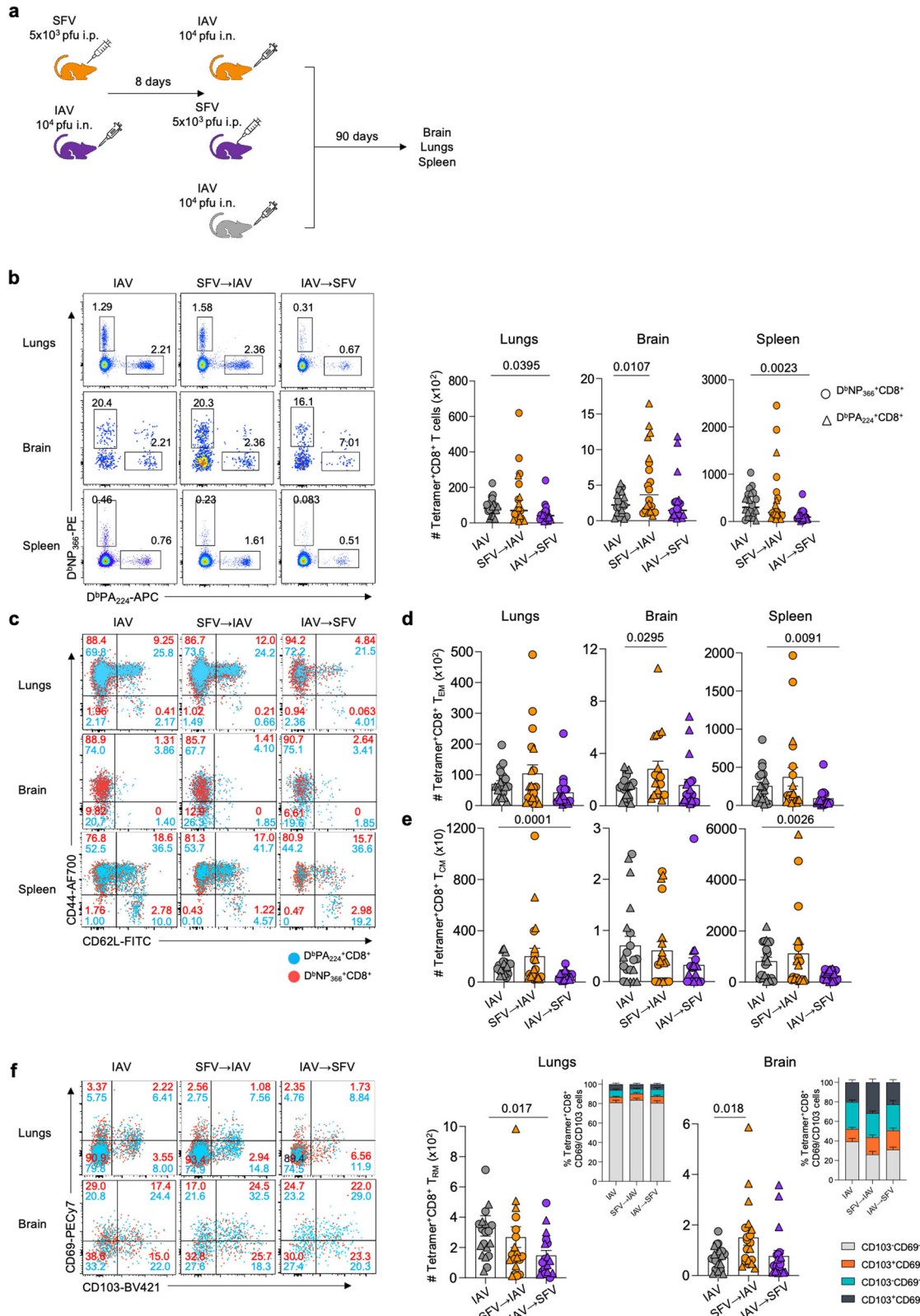

responding to poorly presented antigens, such as $D^bPA_{224}$, lack robust proliferative capacity upon recall[63]. Therefore, it appears that in IAV→SFV sequential infection, SFV interfered with IAV antigen presentation during the late phase of the primary IAV response by impeding the ability of DCs to present IAV antigens and overwhelming the system with SFV antigens. $D^bNP_{366}$ is efficiently presented during the late phase of the primary response by virtue of prolonged

engagement with DCs[63], this interaction could have been impacted by outcompeting abundant SFV antigens. As a result, $D^bNP_{366}$ and $D^bPA_{224}$ were effectively presented only during the early phase of the primary IAV-specific CD8+ T cell response, which resulted in the abolition of immunodominance hierarchy.

Altogether, as demonstrated by our study, viral sequential infection can alter the magnitude, function and trafficking of CD8+ T cell

**Fig. 7 | Impaired immunological memory formation following sequential infection. a** Mice were infected i.p. with $5 \times 10^3$ pfu A7(74) SFV followed 8 days later with $10^4$ pfu A/HKx31 (i.n.). Another group was infected with the same virus dosage, but in reverse order. IAV-only group has been included as control. Brains, lungs and spleens were harvested on d90. **b** Absolute numbers of IAV-specific CD8+ T cells directed at both $D^bNP_{366}$ and $D^bA_{224}$ epitopes are shown across different anatomical sites in IAV, SFV→IAV and IAV→SFV infected mice. **c–e** Concatenated FACS plots ($n = 5$) shown for each group and tissue. Absolute numbers of IAV-specific $T_{EM}$ and $T_{CM}$ CD8+ T cells are shown for both $D^bNP_{366}$+CD8+ T cell and $D^bPA_{224}$+CD8+ T cell specificities in brain and lungs of IAV, SFV→IAV and IAV→SFV infected mice. Representative FACS plots shown for each group and tissue ($n = 10$, error bar represents SEM). **f** Absolute numbers of IAV-specific $T_{RM}$ CD8+ T cells for both $D^bNP_{366}$+CD8+ and $D^bPA_{224}$+CD8+ T cells in brain and lungs of IAV, SFV→IAV and IAV→SFV infected mice. Insets show frequencies of CD8+ T cells gated on CD69 and CD103 ($n = 10$, error bar represents SEM). Concatenated FACS plots ($n = 5$) shown for each group and tissue. Gating strategy is shown in Supplementary Fig. S5c. Significance was determined by unpaired two-tailed Student's $t$ test. Source data are provided as a Source Data file.

responses to unrelated viruses, both in acute and memory phase. It is striking that a neurotropic virus affecting completely different tissue such as brain has a profound effect on respiratory immunity. An important consideration in SFV/IAV co-infection is the fact that majority of neurotropic infections are asymptomatic and not being considered a risk factor during an influenza virus infection. Yet, prior arbovirus infection can greatly affect the outcome of IAV-specific immune responses, resulting in a severe respiratory disease or compromised long-term immunity.

## Methods

### Animals and ethics statement

C57BL/6 mice were bred at the Peter Doherty Institute Bioresources Facilities. Animal experiments followed the NHMRC Code of Practice for the Care and Use of Animals for Scientific Purposes guidelines and were approved by the University of Melbourne Animal Ethics Committee (AEC 1714184 and 21319). All mice were monitored daily for clinical signs including determination of their body weight. All animals were kept in HEPA filtered, individually ventilated cages, with environmental enrichment, 12 h light dark cycle and food and water *ad libitum*. Tissues from infected animals were processed in class II BSC.

### Viruses, infection, OT-I system, and CpG administration

In all, 6–8 weeks old male and female mice were initially infected with avirulent A7(74) strain of Semliki Forest virus[64]. All mice were inoculated intraperitoneally (i.p.) with $5 \times 10^3$ pfu of A7(74) in 0.1 mL PBS containing 0.75% bovine serum albumin (PBSA). For subsequent influenza A virus (IAV) infections, mice were lightly anaesthetised with isoflurane and infected by intranasal (i.n.) instillation (30 µL) with $10^4$ pfu of A/HK/x31 (X31; H3N2). For the secondary IAV infection experiments, mice were challenged with $10^3$ pfu A/PR/8/34 i.n. (PR8, H1N1) at ~100 days following the primary IAV infection. To investigate immune mechanisms using the OT-I system, $1 \times 10^6$ of FACS-sorted CD8+CD44loCD62LhiCD45.1+ OT-I cells[65] were transferred i.v. into the C57BL/6 mice. Mice were subsequently infected with $10^4$ pfu x31-OVA recombinant virus[34] 1 day later. In selected experiments, mice were administered i.p. with 20 nmol CpG 1668 (Bioneer, Oakland, CA, USA) to induce aseptic inflammatory response.

### Tissue sampling and cell preparation

Brain, lungs, spleen and mediastinal lymph nodes (MLN) were collected from mice at various time points after infection. To remove blood from the tissue vasculature, following terminal anaesthesia, animals were perfused with 10 mL PBS through the left cardiac ventricle. The comparison of transcardiac perfusion with intravascular antibody labelling demonstrated that perfusion is sufficient to analyse cell populations in the brain, including naïve brain[66]. Brains were removed and processed for virus infectivity assay and chemokines/cytokines levels (half brain bisected sagittally along the midline), preparation of RNA for gene expression studies (half brain), analysis of cell infiltrates (entire brain), or histopathological assessment (entire brain). Single-cell suspensions were purified from the brain, spleen, lungs and mediastinal lymph nodes (MLN). The brain tissue was digested with 1785 units/mL collagenase type III (Worthington Biochemical Corporation, Lakewood, NJ, USA) and 6 units/ml DNase I (Sigma-Aldrich,

St. Louis, MO, USA). CNS-infiltrating leucocytes were isolated from the brain samples by centrifugation on a Percoll gradient (70%, 37% and 30% Percoll). Lungs were either homogenised and centrifuged to obtain clarified supernatants to assay for viral titres/cytokine composition or enzymatically digested in 1 mg/mL collagenase III (Worthington) and 0.5 mg/mL DNase I (Sigma-Aldrich) before passing through cell sieves to obtain single-cell suspensions for analysis. Where necessary, cell suspensions from tissues were incubated with 0.15 M NH4Cl and 17 mM Tris-HCI at pH 7.2 for 5 min at 37 °C to lyse red blood cells.

### Measurement of viral loads

Plaque assay on Vero cells (ATCC, #CCL-81) was used to titrate infectious SFV[13]. Vero cells were seeded in 12 well plates ($2 \times 10^5$ cell per well) and incubated overnight at 37ºC in a 5% $CO_2$ incubator in DMEM media (Life Technologies) containing 10% FCS, 200 mM Glutamax (Life Technologies), and 1% Penicillin/Streptomycin (Life Technologies). Following incubation cells were washed and infected with 200 µL of 10-fold dilutions ($10^{-1}$ to $10^{-8}$) of the brain homogenate in DMEM at 37 °C for 1 h followed by 1 ml overlay medium (1% FCS and 1% HEPES and 0.5% Glutamax and 1% Penicillin/Streptomycin and 1.6% w/v carboxy cellulose), and incubated for 72 hr at 37ºC. Cells were fixed with 1 ml of 10% normal formalin for 1 hr and stained with 0.5% crystal violet (15 min) to visualise the plaques. All dilutions were assayed in duplicates. Titres of infectious IAV in the lungs were determined by plaque assays on confluent Madin-Darby canine kidney (MDCK) cell (ATCC, #CCL-34) monolayers in six-well plates[57]. Cells were infected with lung homogenates at 10-fold dilutions for 45 min at 37 °C before the addition of Leibovitz L15 or MEM medium containing 0.9% agarose overlay containing Trypsin (Worthington Biochemical). Plates were incubated at 37 °C 5%CO2 for 3 days and plaques counted on the cell layer and expressed as plaque forming units (PFU).

### Identification of SFV immunodominant peptide

A7(74) SFV genome was analysed in all three reading frames using peptide prediction websites (http://tools.iedb.org/mhci/, http://www.cbs.dtu.dk/services/HLArestrictor/)[67] and peptide sequences were generated based on binding affinity prediction to mouse MHC-I (H2-K^b and H2-D^b). Peptides (GenScript USA Inc) were resuspended up to 1 mM with DMSO, aliquoted and stored at −20 °C. Enriched T cell populations from brain and spleen were stimulated with 1 µM of the selected peptide for 5 hrs at 37 °C, 5% $CO_2$ in the presence of 1000 U/mL recombinant IL-2 (Roche, Basel, Switzerland) and 1 µL/mL Golgi-Plug (BD Biosciences, San Jose, CA, USA) as described[44]. Cells were washed and stained with CD8-PerCP Cy5.5 (clone 53-67, BD Biosciences 551162) for 30 mins on ice, fixed, permeabilised and stained for cytokines (IFN-γ-FITC, TNF-α-APC and IL-2-PE (Biolegend, San Diego, CA, USA)). Samples were acquired using BD Canto II, and the total cytokine production was calculated by subtracting background fluorescence using no peptide controls. PMA/ionomycin stimulated cells were used as positive controls. The known sequence of the peptides giving positive results were blasted against the NCBI database (https://blast.ncbi.nlm.nih.gov/Blast.cgi) to determine the position of the peptide in SFV proteome.

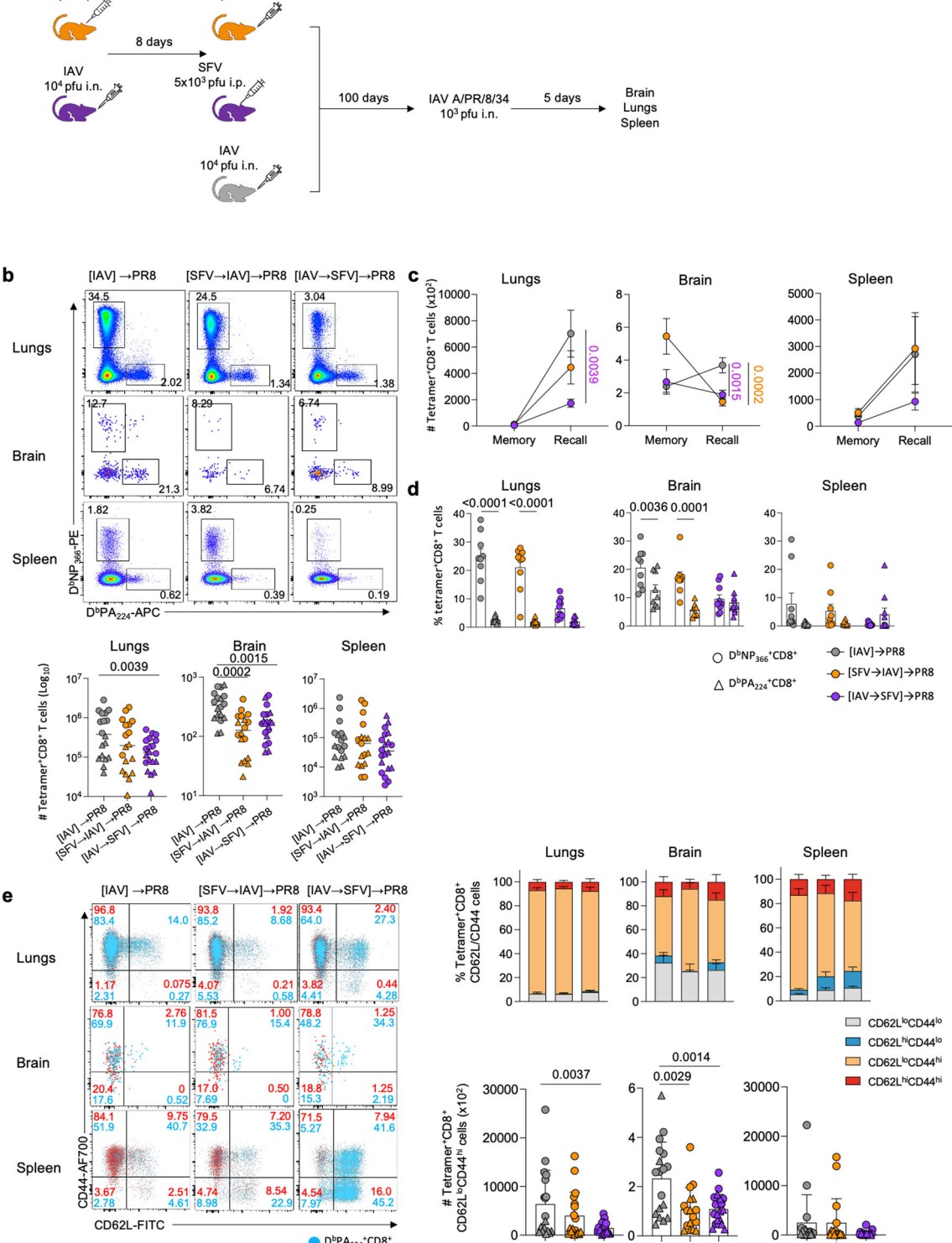

## Tetramer and immunophenotypic staining

MHC-I tetramers targeting the immunodominant epitope of the influenza nucleoprotein (D$^b$NP$_{366-374}$: ASNENMETM; D$^b$PA$_{224-233}$: SSLENFRAYV) and SFV envelope protein (K$^b$-E1$_{159-166}$: TQFIFGPL) were produced in-house and conjugated to streptavidin-PE (BD Biosciences 554061), BV421 (BD Biosciences 563259) and APC (BD Biosciences 554067) respectively at 1:200 dilution at room temperature for 1 h

in the dark. Lymphocytes were stained with combinations of fluorochrome-conjugated antibodies: BD Biosciences: anti-CD8α-PerCyP Cy5.5 (53-67; 551162; 1:200), anti-CD44-Alexa Flour 700 (1M7; 560567; 1:200), anti-CD4-APC Cy7 (GK1.5; 552051; 1:500), anti-TCRβ-BV711 (H57-597; 563135; 1:200), anti-CD25-PECF594 (PC61; 562694; 1:200), anti-CD45.1-FITC (A20; 561871; 1:800), anti-CD62L-FITC (MEL-14; 561917; 1:200), anti-CD45.1-PE (A20; 553776; 1:200), anti-V2αTCR-FITC

**Fig. 8 | Abolition of immunodominance hierarchy in sequentially infected mice. a** Mice were infected i.p. with $5 \times 10^3$ pfu A7(74) SFV followed 8 days later with $10^4$ pfu A/HKx31 (i.n.). Another group was infected with the same virus dosage, but in reverse order. IAV-only group has been included as control. On d100 after primary IAV infection mice were infected i.n. with $10^3$ pfu A/PR/8/34 (PR8). Brains, lungs and spleens were harvested on d5 following PR8 challenge. **b** Absolute numbers of $D^bNP_{366}{}^+CD8^+$ and $D^bPA_{224}{}^+CD8^+$ T cells across different anatomical sites in [IAV]→PR8 ($n = 9$), [SFV→IAV]→PR8 ($n = 9$) and [IAV→SFV]→PR8 ($n = 10$) infected mice are shown. Representative FACS plots shown for each group and tissue, error bar represents SEM. **c** Magnitude of the secondary IAV-specific responses upon rechallenge. Memory: IAV ($n = 10$), SFV→IAV ($n = 10$) and IAV→SFV ($n = 10$). Recall: [IAV]→PR8 ($n = 9$), [SFV→IAV]→PR8 ($n = 9$) and [IAV→SFV]→PR8 ($n = 10$). Mean values plotted, error bar represents SEM. **d** Proportion of IAV-specific $D^bNP_{366}{}^+CD8^+$ T cells and $D^bPA_{224}{}^+CD8^+$ T cells across different anatomical sites in [IAV]→PR8 ($n = 9$), [SFV→IAV]→PR8 ($n = 9$) and [IAV→SFV]→PR8 ($n = 10$) infected mice. Mean values plotted, error bar represents SEM. **e** Absolute numbers and frequencies of IAV-specific CD8$^+$ effector T cells (CD62L$^{lo}$CD44$^{hi}$) for $D^bNP_{366}{}^+CD8^+$ and $D^bPA_{224}{}^+CD8^+$ T cells across different anatomical sites in [IAV]→PR8 ($n = 9$), [SFV→IAV]→PR8 ($n = 9$) and [IAV→SFV]→PR8 ($n = 10$) infected mice are shown. Concatenated FACS plots ($n = 5$ from one experiment) shown for each group and tissue, error bar represents SEM. Gating strategy is shown in Supplementary Fig. S5c. Significance was determined by ordinary two-way ANOVA test with Šídák's multiple comparison. Source data are provided as a Source Data file.

(B20.1; 553288; 1:100), anti-CD45R-APCCy7 (RA3-6B2, 552094; 1:200), anti-CD38-BV711 (90; 740697; 1:200), anti-CD138-PE (281-2; 553714; 1:400), anti-CD11c-FITC (HL3; 557400; 1:1000), anti-Gr1-FITC (RB6-8C5; 553126; 1:400), anti-CD64-AF647 (X54-5/7.1; 558539; 1:150), anti-Ly6C-AlexaFlour700 (AL-21; 561237; 1:400), anti-CD11b-BV605 (M1/70; 563015; 1:600), anti-CD45.2-BV711 (104; 563685; 1:400), anti-CD11c-PE (HL3; 553802; 1:400), anti-SigLecF-PECF594 (E50-2440; 562757; 1:1200), anti-Ly6G-PECy7 (1A8; 560601; 1:1200).

BioLegend: anti-CD62L-BV570 (MEL-14; 104433; 1:200), anti-CD279-BV785 (29 F.1A12; 135225; 1:200), anti-CD38-PECy7 (90; 102718; 1:500), anti-CD8α-BV510 (53-6.7; 100752; 1:200), anti-CD103-BV421 (2E7; 121422; 1:200), anti-CD69-PECy7 (H1.2F3; 104512; 1:200), anti-CD62L-PECy7 (MEL-14; 104418; 1:200), anti-GL7-PerCyPCy5.5 (GL7; 144610; 1:200), anti-CD19-APC (6D5; 11512; 1:400), anti-I-Ab-PacBlue (AF6-120.1, 116422; 1:400), anti-IgD-PECy7 (11-26 c.2a; 405720; 1:200), anti-CD3-FITC (145-2C11; 100306; 1:400), anti-F4/80-FITC (RB6-8C5; 553126; 1:150).

Invitrogen eBiosciences (anti-KLRG1-FITC (2F1; 11-5893-82; 1:200). Cell viability was determined by staining with either Live/Dead-Aqua 525 (L34966A, ThermoFisher, 1:800) or Live/Dead Fixable Near-IR (L10119, ThermoFisher, 1:800). Cells were fixed with 1% paraformaldehyde before analysis by flow cytometry. Antibody staining was performed at 4 °C in the dark. Samples were subsequently acquired on a BD LSR Fortessa (BD Biosciences) flow cytometer and data analysed by FlowJo Software (Tree Star Inc., USA). Gating strategies for flow cytometry data are shown in Supplementary Fig. 5.

**Cytokine and chemokine analysis**

Cytokine and chemokine levels in tissue homogenates were analysed using the LEGENDPlex Multi-Analyte Flow Assay Kit (Biolegend) Mouse Anti-Virus Response Panel (13-plex) according to manufacturers' instructions.

**Isolation and adoptive transfer of OT-I cells**

Naïve CD45.1$^+$ OT-I cells isolated from OT-I/Ly5.1 TCR transgenic mice were purified from single-cell suspensions prepared from lymph nodes (LNs). OT-I cells were FACS-sorted based on the expression of surface markers (CD8$^+$CD44$^{low}$CD62L$^{hi}$CD45.1$^+$TCRVα2$^+$) using a BD Aria III (BD Biosciences). Mice received 1x10$^6$ CD45.1$^+$ OT-I cells intravenously in a volume of 200µl. Purified CD45.1$^+$ OT-I cells were activated in vitro with K$^b$-restricted OVA$_{257-264}$ Class I (SIINFEKL) peptide-pulsed splenocytes as described[68]. Mice were injected intravenously with 1x10$^6$ cells/mouse.

**In vivo T cell proliferation assay**

VPD450 (Violet Proliferation Dye)-labelled OT-1 cells (1x10$^6$) were transferred intravenously into C57BL/6 mice on day 7 post-SFV infection, or day −1 for X31-OVA infection. One day later, mice were inoculated with recombinant X31 influenza virus containing the K$^b$OVA$_{257-264}$ epitope (amino acid sequence SIINFEKL) within the stalk of the NA protein (X31-OVA)[34,35]. Mediastinal lymph nodes, lungs, and spleen were obtained on day 3.5 and 4.5. VPD450 levels of CD8$^+$CD45.1$^+$ cells were assessed by flow cytometry.

**RT-PCR**

Half brain specimens were submerged in RNA stabilisation reagent, RNAlater (Qiagen, Hilden, Germany). RNA was extracted using the RNeasy Lipid Tissue Mini kit (Qiagen) according to manufacturer's instructions and stored at −80°C until use. RNA quantity was determined by spectrophotometry (NanoDrop 2000, Thermo Fisher Scientific, Waltham, MA, USA). 5 µg of RNA was converted to cDNA using SuperScript III reverse transcriptase (Invitrogen, Waltham, MA, USA). RT-PCR was carried out using Fast SYBR Green Mastermix (Thermo Fisher Scientific) and primers recognising the NSP3 region of SFV or Matrix (M) region IAV genomic RNA: viral NSP3 reverse primer- 5′-GGGAAAAGATGAGCAAACCA-3′; viral NSP3 forward primer- 5′-GCAA-GAGGCAAACGAACAGA-3′; viral PR8 M reverse primer- 5′-GGGCAT-TYTGGACAAAKCGTCTACG-3′; viral PR8 M forward primer – 5′-GACCRATCCTGTCACCTCTGAC-3′. Levels of virus RNA were normalised to the housekeeping gene, GAPDH: mouse GAPDH reverse primer- 5′ TTGTCAAGCTCATTTCCTGGT-3′; mouse GAPDH forward primer – 5′ TTACTCCTTGGAGGCCATGTA-3′. Samples were run on a QuanStudio 7 Real time PCR machine (ThermoFisher) with the following PCR conditions: the first cycle was 5 min at 95 °C, 10 sec at 95 °C, 15 sec at 60 °C and 1 sec at 72 °C. The following 45 cycles: 10 sec at 95 °C, 15 sec at 60 °C, 1 sec at 72 °C and ended with 10 min at 40 °C. The relative amount of virus genomic RNA was calculated using the $2^{-\Delta\Delta CT}$ method.

**Single-cell sorting and TCRαβ analysis**

Following staining, single $D^bNP_{366}{}^+CD8^+$ and $D^bPA_{224}{}^+CD8^+$ T cell populations were indexed single-cell sorted on a BD FACSAria III for TCR analysis performed as described[21,24]. Multiplex-nested RT-PCR-amplified CDR3α and CDR3β regions from single cells[69,70] were analyzed by IMGT/V-QUEST before pairing through the TCRdist pipeline for modelling amino acid motifs, TCR landscapes, neighbour distance distribution and probabilities of generation (P$_{gen}$)[71]. Testing for variations in P$_{gen}$ across epitope specificities was performed essentially as described[24] using linear mixed models[71]. Data visualisation for CIRCOS plots was performed in R using a package for circular layout[72].

**Determining blood-brain barrier permeability**

Blood-brain barrier (BBB) permeability was examined by an Evans Blue extravasation method[29,30]. Evans blue dye (EBD) (Sigma Aldrich) was prepared as a 2% solution in PBS and injected as a single bolus dose of 4 ml/kg intraperitoneally. To enable EBD's uniform blood distribution mice were culled 3 hours after injection. Animals were perfused with PBS through the left cardiac ventricle to remove blood from the brain vasculature, brains were removed and placed in 1:3 weight (mg):volume (ml) ratios of 50% trichloroacetic acid (TCA), then homogenised. TCA extracts from the brain samples were centrifuged at 10,000×$g$ for 20 min at 4°C to remove precipitates and tissue debris, and the supernatants were added to a 96-well plate at 30 µl per well, each plate supplemented with 90 µl of 95% ethanol and thoroughly mixed by

pipetting. Fluorescence intensity was measured at 610 nm excitation/680 nm emission using the CLARIOstar Plus machine (BMG Labtech, Ortenberg, Hessen, Germany). The amount of EBD recovered from brains was expressed as micrograms of dye per gram of tissue using 8-point standard curve after correcting for the background of supernatants obtained from brains of uninfected mice.

## LEGENDscreen and InfinityFlow analysis

Single-cell suspension of brains, lungs, and spleens of single and co-infected mice ($n = 16$) were enriched for CD8$^+$ T cells using magnetic-activated cell sorting (MACS) CD8 enrichment kit (Miltenyi Biotec) following the manufacturer's instructions. Six samples were then bio-tin labelled with anti-mouse CD45.1 (A20, BioLegend 110704), followed by combinational barcoding with streptavidin BUV396 (BD Biosciences 564176), BUV496(BD Biosciences 612961), and BV421 (BD Biosciences 563259). Brain CD8$^+$ T cells of IAV-infected mice were BUV395 labelled, brain CD8$^+$ T cells of SFV→IAV infected mice were BUV 496 labelled, lung CD8$^+$ T cells of IAV-infected mice were BV421 labelled, lung CD8$^+$ T cells of SFV→IAV infected mice were BUV 395 + BUV 496 labelled, spleen CD8$^+$ T cells of IAV-infected mice were BUV395 + BUV421 labelled, and spleen CD8$^+$ T cells of SFV→IAV infected mice were BUV496 + BV421 labelled. After barcoding, samples were stained with anti-CD8a, KLRG1, CD44, CD38, PD-1 and CD62L antibodies as a backbone panel, washed, pooled, and aliquoted to 75,000 cells per well in assay plates. Preparation of assay plates, staining, and fixation was performed according to the manufacturer's protocol (LEGENDScreen Mouse PE Kit, BioLegend). Following data collection, FCS files were examined in FlowJo (Tree Star Inc., USA) for quality control, and CD8$^+$ T cells were exported into new FCS files for each well to be used as input for InfinityFlow prediction pipeline using the infinityFlow R package as described in ref. 38. Specifically, 10,000 events randomly selected from each file to train XGBoost non-linear regression model implemented in the xgboost R package with a depth of 500 trees, a learning rate of 0.05 and otherwise default parameters. This model aimed at predicting the intensity of the PE-conjugated variable antibody expression from the backbone expression data for each cell in the well. From non-training data, up to 10,000 events were randomly selected for each well and concatenated across wells to generate the backbone matrix of the InfinityFlow output. Predictions from all models were generated across all of these non-training events to generate the final InfinityFlow matrix, containing backbone-measured and variable-predicted expression intensities. These predictions were finally concatenated and dimensionally reduced with UMAP on backbone parameters to generate a single data matrix. For downstream analysis, OT-I T cells originating from distinct organs and infection contexts were then debarcoded and analyzed with the OMIQ cloud platform (omiq.ai). InfinityFlow augmented data matrix was scaled using hyperbolic arcsine (asinh) transformation and UMAP dimensionality reduction using InfinityFlow prediction was performed with the following parameters: n_neighbors = 15, min_dist = 0.1, metric = euclidean and n_epocs = 200. Clustering was performed using FlowSOM with InfinityFlow predicted markers in OMIQ.

## Histopathology

Lungs were fixed in 4% paraformaldehyde in PBS, processed for embedding in paraffin wax, cut into thin sections, stained with haematoxylin and eosin, and analysed microscopically. Multiple sections were cut through the lungs at 5 μm thickness at 100 μm intervals (x5 levels).

Histopathology was performed by the Phenomics Australia Histopathology and Digital Slide Service at the University of Melbourne.

## Statistical analyses

Statistical analyses were performed using Student's unpaired $t$ test within GraphPad Prism 9 software.

## Reporting summary

Further information on research design is available in the Nature Portfolio Reporting Summary linked to this article.

## Data availability

All data generated or analysed during this study are included in this published article (and its supplementary information files). Source data are provided with this paper as Source Date file 'Source data file Foo et al.xlsx'. The UMAP data was generated using InfinityFlow predicted expression values ($n = 255$ markers). Raw fcs files and concatenated InfinityFlow predictions are provided as 'LegendScreen UMAP data.zip' and 'LegendScreen raw fcs files.zip' via Figshare: https://doi.org/10.6084/m9.figshare.25332556. All relevant data are also available from the authors. The TCR sequences that support the findings of this study are available on the Mendeley database https://data.mendeley.com/datasets/6xb4j8xtv5/1 (https://doi.org/10.17632/6xb4j8xtv5.1), are supplied in Supplementary Tables 1-3 and a separate file 'TCR sequences analysis.xlsx'. Source data are provided with this paper.

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

## Acknowledgements

We acknowledge the Melbourne Cytometry Platform (The Doherty Institute) for provision of flow cytometry services and Phenomics Australia Histopathology and Digital Slide Service at the University of Melbourne for the histopathological assessments. We would like to thank Melbourne Bioresources Platform staff at the Doherty Institute. The work was funded by a University of Melbourne grant to J.K.F., the NHMRC Leadership Investigator Grant to K.K. (#1173871) and J.A.V. (#2016969), an NHMRC Project Grant (#1163090) and an NHMRC Principal Research Fellowship (#1154502) to J.A.V.; I.J.H.F. is a recipient of the Melbourne Research Scholarship. X.J. was a recipient of the China Scholarship Council-UoM Joint Scholarship. K.K. is NHMRC L1 Investigator Fellow (#1173871) and University of Melbourne Dame Kate Campbell Fellow, EBC is NHMRC Peter Doherty Fellow (#1091516), J.A.V. is NHMRC L3 Investigator Fellow.

## Author contributions

L.K., K.K., and J.K.F. supervised the study. L.K., K.K., J.K.F., and I.J.H.F. designed the experiments. I.J.H.F., L.K., B.Y.C., E.B.C., S.Y.C., X.J., A.H.Y.Y., and A.F.C. performed and analysed experiments. H.A.M. and M.E. analysed data. M.A., H.E.G.M., and L.M.W. provided reagents. M.A., H.E.G.M., J.A.V., M.E., L.K.M., and L.M.W. provided intellectual input into the study design and data interpretation. L.K., K.K., J.K.F., and I.J.H.F. wrote the manuscript. All authors reviewed and approved the manuscript.

## Competing interests

The authors declare no competing interests.
