## [Peer Review File · Nature Communications]

REVIEWER COMMENTS

Reviewer expertise:

#1 - immunity, influenza, mouse model

#2 - T cells, immune response to emerging pathogens

#3 - neuroinflammation, CNS, animal models

Reviewer #1 (Remarks to the Author):

Immune responses to viral infections are typically studied in the context of single infections or as co-infections with bacteria. In reality co-infections with two viruses do occur regularly but these are rarely investigated. So our understanding of the impact of viral-coinfections on immune responses is very limited. The paper submitted by Foo and colleagues investigates how infection with Semliki Forest virus, which typically evokes only mild symptoms in humans impacts subsequent influenza A virus infection. They observed that a SFV infection reduces viral clearance and increases pathology of a subsequent influenza A virus infection. This was associated with increased trafficking of antigen-specific CD8 T cells to the brain but with reduced infiltration of these cells in the lungs. Furthermore, SFV infection reduced CD8 T cell proliferation and memory formation. This could be attributed to paralysis of APCs by the SFV infection. However, this dysregulated T cell activation and memory formation did not impact recall of antigen-specific CD8 T cells upon a subsequent infection with a heterosubtypic IAV infection. Surprisingly, a SFV infection following the first IAV infection strongly reduced CD8 T cell recall upon the secondary infection. This is a very important finding demonstrating how co-infection might have an important impact on the shaping of immunity after viral infections and possibly also vaccination. The findings of Foo et al are highly relevant and supported by well-designed experiments and in depth analysis. I have no major but some minor comments.

1. Figure 1 demonstrates that a SFV infection does not impact the presence/infiltration of various leukocytes (not considering T cells there yet) upon IAV infection. Figure 2 starts with exemplary FACS plots showing that SFV infection evokes infiltration of CD8 T cells in the brain but has no impact on the overall CD8 T cell infiltration in the lungs. However, SFV has a clear effect on the infiltration of antigen-specific CD8 T cells in the lungs. For the narrative of the report I would suggest to include the overall infiltration of CD4 and CD8 cells in the last panel of figure 1 and show the defect in antigen-specific CD8 T cell infiltration in the lungs in figure 2. This is merely a suggestion, but I would highly recommend to include the CD4 cells as part of the last panel of Fig. 1.

2. The report focusses on the induction of CD8 T cells. Did the SFV infection also impact the IAV antibody response? Was serum collected before the secondary IAV infection in the rechallenge experiment? Please test if SFV infection impacted the induction of X31 neutralizing antibodies and

PR8 cross-binding antibodies (eg using PR8 infected cells in FACS analysis). In addition, were antigen-specific CD4 cells investigated as well?

3. Please specify the LD50 level of the used X31 virus inoculum. Was there bodyweight loss upon IAV infection in figure 1. If so, please include these data.

4. Keeping in mind the presence of IAV RNA in the brain and the antigen specific recruitment of OT-I cells to the brain of IAV-OVA infected mice, was the presence of viral proteins in the brain tested via immunochemistry?

5. In mice IAV infection induces heterosubtypic immunity via CD8 T cells. Please specify in the rechallenge experiment whether the X31 infection provided any protection against the subsequent PR8 infection. In this experiment no control group was included that did not receive the first IAV infection for monitoring heterosubtypic immunity. So to get some notion of the level of cross-protection, please specify the LD50 of the PR8 inoculum and include the bodyweight of the mice after the primary and secondary IAV infection. Was the impact of a subsequent SFV infection on the IAV CD8 T cell recall associated with differences in bodyweight?

Reviewer #2 (Remarks to the Author):

In this study by Foo et al., the researchers investigate changes to the immune response when a mouse is co-infected with two unrelated viruses, Influenza A virus and Semliki Forest virus. This is an interesting topic of potential great importance. They found that mice that were first infected with SFV then IAV has increased lung pathology, exacerbated disease, increased T resident memory in brain tissue, and change in IAV specific memory clonotypes. The authors also investigate if the trafficking of antigen specific cells requires antigen to be present in the brain tissue or if the trafficking is due to prior infection with SFV permeabilizing the blood brain barrier. The researchers explored if the suboptimal clonotypes were caused by APC paralysis by SFV virus. Overall, they found that pre-infection with SFV leads to increased BBB permeabilization and allows for increased presence of IAV specific T cells in the brain. They also found that pre-infection increased IAV-mediated pathology. While the significance of the research and the data collected are clear there are some concerns that somewhat dampen enthusiasm for the paper.

Major Concerns

-One major concern is that the majority of the experiments are lacking the key control of infection with SFV alone. This should be provided to be able to really put their findings in the context of co-infection vs single infections.

-The timeline for the study is also somewhat unclear. Why day 8 following the first-infection? Are there longer-term impacts of SFV infection? This was only tested in the context of previous IAV infection rather than SFV and rechallenge with IAV at memory time points?

- The number of SFV specific CD8 T cells in the lung should be investigated. Is this responsible for the increased pathology?

Minor Concerns

- The conclusions regarding viral clearance are overstated. The data shows that on day 7 post infection neither group cleared the virus completely, therefore, they can only state at day 7 there was a higher viral load in the co-infected mice. They would need more timepoints to state if one group cleared the virus faster than the other. This should be reworded

- Similarly, the cytokine data shows that there are still both tissue and circulating levels of cytokines in both the control and co-infected mice. While the levels of cytokine are higher in the co-infected mice, they are not absent in the control mice. The author should dampen the language that the cytokine storm is prolonged in the co-infected mice without showing data from more days.

- In the IAV trafficking to the brain section, this sentence is confusing:

o "After 3 hrs, mice were culled, the brain vasculature perfused, and brain dye levels determined. distributed throughout the body for 3 hr prior to culling mice and brain perfusion."

- Did they use in vivo labelling to differentiate cells in the vasculature from cells in the tissue prior to identifying Trm? This is much more accurate than perfusion of the tissue.

- Authors should state which clone of CD43 they used and if clone 1b11 is used it should be stated clearly that they are measuring changes in the glycosylation of CD43 rather than changes in CD43 expression.

- The presence of CD8 T cells in the brain following IAV infection alone should be at least discussed

- Fig 1b. What is the line connecting the 2 sets of dots curved? There is no data to support changes one way of the other between time points.

- Figure 2c. Numbers should be provided here along with proportions. Could this be complemented with a tetramer analysis?

- The dot plots in figure 7c, 7f, and 8e are very difficult to interpret. The authors should not overlay the representative graphs.

- Some of the discussion is overstated and needs to be revised.

Reviewer #3 (Remarks to the Author):

This manuscript reports the effect of prior SFV infection on IAV-specific anti-viral immune responses using C57BL/6 mouse models to address a larger question regarding potential differences in immune mechanisms between single and co- viral infections. This is an important question, however, the study design and investigation are limited and skewed to look at the question in a forward manner, without consideration of how the 2nd infection may impact the 1st and potentially contribute to the immunological changes observed. The inclusion of SFV, which is rarely found to infect humans and when found, tends to be much milder than in rodents, is not clear from a translational perspective. The authors should provide some rationale for the inclusion of SFV, rather than a different neurotropic virus that is translatable to a potential human condition. Although two viruses are used, the study is focused on the response to IAV, alone, or following SFV at 8-days post-SFV infection. This investigation revealed elevated cytokine production and greater TCR heterogeneity with dual infection, which is not unexpected. The absence of an SFV only group is disappointing. As the authors note, i.p. inoculation of SFV results in systemic infection. The potential for extracerebral SFV infection has been reported with i.p. inoculation and should be included in this study. The potential for IAV exacerbating SFV injury and/or virus replication should also be investigated, as it is reasonable to anticipate the SFV is not fully cleared at the time of IAV exposure. According to Figure 6, it does appear that SFV is in lung at day 3 post-IAV infection but is unclear how long this persists. It is reasonable to expect that SFV does not get cleared and is presumably still present in the CNS. This may also provide some insight into the IAV kinetics in lung and/or other reported findings, including cytokine levels, lung pathology, virus distribution, and the reduced numbers of DbNP366+CD8+ and DbPA224+CD8+ T cells in the lung and spleen. Pathology in all major organ systems should be investigated through histopathological techniques, including thorough investigation into organ distribution of both viruses and cell-type analyses. Due to the route of SFV infection, gut is a significant system that should also be included.

IAV infection is at 8-days post-SFV infection, which has not provided sufficient time for the mice to even partially recover from CNS injury caused by SFV. As such, it is not surprising that IAV-specific CD8+T cells would be present in the CNS, which are 'called-in' by CNS-mediate chemokines because of the on-going inflammation in the brain. This really isn't a 'redirection', as suggested by the authors, but rather the immunological consequence of two injured organ systems eliciting T-cell help. The brain should be investigated for pathology and persistent SFV, particularly since only a subset of animals were demonstrated to have IAV-specific CD8 T-cells, which may suggest some animals cleared SFV or have less severe CNS pathology than others. It is not clear why the OT-I method was used when the brain can be investigated specifically for SFV. Reduced BBB integrity is not unexpected with either virus infection but does not account for leukocyte infiltration into the

CNS, which can be seen easily by H&E. Importantly, did any SFV infected animals experience encephalitis? Please provide the H&E images of the brain that are described in the Methods section.

The Methods section states that SFV in brain was assessed with Vero cells, however, this is only shown in lung.

Please include tissue thickness in histopathological methods within the Methods section.

RESPONSES TO REVIEWERS' COMMENTS:

We thank the Reviewers for their comments and insightful suggestions. We acknowledge that in addressing them, we have improved the manuscript. We have provided a point-by-point response below.

We also thank the *Nature Communications* Editors for giving us the opportunity to resubmit the modified version of our manuscript.

Point-by-point responses to Reviewers' comments:

Reviewer: 1

Comments to the Author

This is a very important finding demonstrating how co-infection might have an important impact on the shaping of immunity after viral infections and possibly also vaccination. The finding of Foo et al are highly relevant and supported by well-designed experiments and in depth analysis.

We thank the Reviewer for the positive feedback and appreciation of our study.

1). For the narrative of the report I would suggest to include the overall infiltration of CD4 and CD8 cells in the last panel of figure 1 and show the defect in antigen-specific CD8 T cell infiltration in the lungs in figure 2. This is merely a suggestion, but I would highly recommend to include the CD4 cells as part of the last panel of Fig. 1.

In addition, were antigen-specific CD4 cells investigated as well?

As per Reviewer's suggestion, we included numbers of infiltrating effector (CD44^{hi}CD62L^{lo}) CD4⁺ T cells as part of revised Figure 1f.

The numbers of infiltrating pan CD4⁺ and CD8⁺ T cells have been included in the Supplementary Figure 1.

We refer to those additional figures in Results (page 6).

Fig. 1f Absolute numbers of B cells (B220⁺CD19⁺), antibody secreting cells (ASC, IgD⁻B220^{lo}CD138⁺), neutrophils (CD11b⁺Ly6G⁺), macrophages (CD11b⁺CD64⁺) DCs (CD11c⁺), inflammatory monocytes (CD11b⁺Ly6C⁺), and effector CD4⁺ T cells (CD62L^{lo}CD44^{hi}CD4⁺) across different anatomical sites (n=14-15, error bar represents SEM).

Fig. S1 Absolute numbers of **b** CD8⁺ T cells and **c** CD4⁺ T cells across different anatomical sites in naïve, SFV, IAV, and SFV→IAV infected mice (n=14-15, error bar represents SEM).

Antigen-specific (IAV-specific) CD4⁺ T cells were not investigated in this study mainly due to poor performance of class II tetramers.

We have included this sentence in the revised manuscript (page 6):

While numbers of activated CD4⁺ T cells mirrored those of activated CD8⁺ T cells, numbers of IAV-specific CD4⁺ T cells could not be investigated due to poor performance of class II tetramers.

2.) Did the SFV infection also impact the IAV antibody response?

Was serum collected before the secondary IAV infection in the rechallenge experiment?

Please test if SFV infection impacted the induction of X31 neutralizing antibodies and PR8 cross-binding antibodies (eg using PR8 infected cells in FACS analysis).

We thank the Reviewer for the insightful comments and agree that antibody responses to influenza could also be affected by prior SFV infection. We performed comprehensive analysis

of B cell compartment by flow cytometry during the acute phase on influenza infection (these data were included in Figure 1 of the original submission). We investigated proportions and numbers of pan B cells (CD19⁺B220⁺), plasma cells (IgD-B220^{lo}CD138⁺) and germinal centre cells (B220⁺CD19⁺CD38^{lo}GL7⁺). We found no differences in lungs, brain and spleen between the groups, therefore we did not follow with further analysis of antibody responses.

Sera were not collected in the rechallenge experiment, therefore, we are unable to test whether induction of X31 neutralising antibodies was impacted by SFV infection.

Nevertheless, it is plausible that although B cell responses did not differ quantitatively, they could differ qualitatively. B cells from co-infected mice could have different activation status, produced antibodies of lower affinity resulting in impaired affinity maturation, and there could have been differences in the rate of class-switching and numbers of neutralising IgM and IgG antibodies, which could further affect seroconversion and seroprotection rates.

Following the Reviewer's comment, we have included the above point in the revised manuscript (page 5). It reads:

Nevertheless, it is plausible that although B cell responses did not differ quantitatively, they could differ qualitatively. B cells from co-infected mice could have different activation status, produced antibodies of lower affinity resulting in impaired affinity maturation, and there could have been differences in the rate of class-switching and numbers of neutralising IgM and IgG antibodies, which could further affect seroconversion and seroprotection rates.

3). Please specify the LD50 level of the used X31 virus inoculum. Was there bodyweight loss upon IAV infection in figure 1. If so, please include these data.

In this study, we used a sublethal dose of X31 virus, therefore LD50 was not used. We used 10⁴ pfu of X31 virus. This has been routinely used in our laboratory to elicit robust immune responses, but not to kill animals or lead to excessive (more than 20% weight loss) body weight loss (Kedzierska *et al.* PNAS 2006 103 (24):9184-9189, Kedzierska *et al.* PNAS 2007 104 (23):9782-9787, Valkenburg *et al.* Nat Communications 2013 4:2663).

Viral infectious doses are listed in the Methods section on page 19.

Body weight loss data have been included in the revised Supplementary Figure 1.

4.) Keeping in mind the presence of IAV RNA in the brain and the antigen specific recruitment of OT-I cells to the brain of IAV-OVA infected mice, was the presence of viral proteins in the brain tested via immunochemistry?

We thank the Reviewer for the suggestion. In our study, we did not investigate the presence of IAV viral proteins in the brain via immunochemistry. It is plausible that some protein could have been detected, but the sensitivity of the plaque assay did not allow for detection of an infectious IAV in the brain, indicating that the amounts of IAV protein in the brain would be very low.

5. In mice IAV infection induces heterosubtypic immunity via CD8 T cells. Please specify in the rechallenge experiment whether the X31 infection provided any protection against the subsequent PR8 infection. In this experiment no control group was included that did not receive the first IAV infection for monitoring heterosubtypic immunity. So to get some notion of the level of cross-protection, please specify the LD50 of the PR8 inoculum and include the bodyweight of the mice after the primary and secondary IAV infection. Was the impact of a subsequent SFV infection on the IAV CD8 T cell recall associated with differences in bodyweight?

We thank the Reviewer for this question. The investigation of heterosubtypic immunity was not the aim of the recall experiment. The rechallenge dose of PR8 was 10^3 pfu delivered intranasally (as indicated above all infectious virus doses are listed in the Methods section as well as throughout the manuscript on page 19). This is a lethal dose in unprimed mice (Ivins *et al.* J Virol, 2017, 91:e00624) and we would expect 100% mortality in this group. Therefore, it was not possible to include PR8-only group in this study due to ethics compliance.

Following Reviewer's suggestion, we included the weight loss in all experimental groups following X31 priming and PR8 rechallenge in the revised Supplementary Figure 4. There were no differences in body weight loss between the groups. Also, no differences were observed in the lung viral load (these data were included in supplementary Figure 4 provided with the initial submission), at least up to day 5 post-PR8 rechallenge (page 13). These results indicate that all experimental groups could mount optimal recall response.

We have included these findings in Results (page 14):

No differences were detected in either weight loss, lung virus titres or cytokine/chemokine milieu between the experimental groups (Figure S4d, S4e and S4f), indicating that all groups could mount optimal recall IAV-specific tetramer⁺CD8⁺ T cell responses against the secondary PR8 IAV infection.

Fig. S4d Weight loss of [IAV]→PR8, [SFV→IAV]→PR8, and [IAV→SFV]→PR8 challenged mice were monitored for 5 days (n= 9-10, error bars represent SEM). **e** Cytokine levels in the lungs were measured by Legendplex, and **f** lung viral loads were determined by plaque assay.

Reviewer: 2

In this study by Foo et al., the researchers investigate changes to the immune response when a mouse is co-infected with two unrelated viruses, Influenza A virus and Semliki Forest virus. This is an interesting topic of potential great importance. They found that mice that were first infected with SFV then IAV has increased lung pathology, exacerbated disease, increased T resident memory in brain tissue, and change in IAV specific memory clonotypes. The authors also investigate if the trafficking of antigen specific cells requires antigen to be present in the brain tissue or if the trafficking is due to prior infection with SFV permeabilizing the blood brain barrier. The researchers explored if the suboptimal clonotypes were caused by APC paralysis by SFV virus. Overall, they found that pre-infection with SFV leads to increased BBB permeabilization and allows for increased presence of IAV specific T cells in the brain. They also found that pre-infection increased IAV-mediated pathology. While the significance of the research and the data collected are clear there are some concerns that somewhat dampen enthusiasm for the paper.

We thank the Reviewer for their feedback and appreciation of our study.

1.) One major concern is that the majority of the experiments are lacking the key control of infection with SFV alone. This should be provided to be able to really put their findings in the context of co-infection vs single infections.

As per Reviewer's suggestion, we have included an SFV alone group in Figure 1 and Figure 2 as well as Supplementary figure 1 of the revised manuscript.

We described these data in the revised manuscript:

On page 5:

SFV-only infection elicited negligible cytokine responses in lung or serum compared to other experimental groups.

*SFV-only infection did not cause detectable histopathological changes in the lungs (**Figure 1d and 1e**).*

*To understand immune responses during sequential SFV → IAV infection, we performed a broad analysis of myeloid and B cell compartments in lungs, spleen and brain on d7 after IAV infection or SFV infection (**Figure 1f**).*

Figure 1 Foo *et al*

2.) *The timeline for the study is also somewhat unclear. Why day 8 following the first-infection? Are there longer-term impacts of SFV infection? This was only tested in the context of previous IAV infection rather than SFV and rechallenge with IAV at memory time points?*

In our study, we have chosen day 8 following the first infection based on the fact that infectious SFV is generally cleared from the brain around 7-8 days post infection (Bradish *et al.* J Gen Virol, 1972, 15:205, Fragkoudis *et al.* Viruses, 2018, 10:273, Kedzierski *et al.* Viruses, 2022, 14:2476), and adaptive immune responses to the primary infection, particularly T cell responses, are reaching their peak around this time.

We agree with the Reviewer that it would be of great interest to test shorter and longer time periods between infections, as we believe there could be long-term effects of SFV infection on IAV immunity. Comprehensively studying the effect of the time interval between initial infection and the subsequent infection on the course of infection is a big undertaking that cannot be included here but we are currently investigating this as a separate project in the laboratory.

Following the Reviewer's comment, we included justification for the d8 secondary infection on page 4 of the revised manuscript:

Day 8 was chosen as adaptive immune responses are at their peak and infectious SFV is generally cleared from the brain around this time^{13, 14}.

3.) *The number of SFV specific CD8 T cells in the lung should be investigated. Is this responsible for the increased pathology?*

We would like to refer the Reviewer to Figure 6a showing the numbers of SFV-specific CD8⁺ T cells. These were assessed on day 7 post primary SFV infection (SFV) or secondary IAV infection (SFV→IAV) at the same timepoint as the histopathology in Figure 1. There is no difference in SFV-specific CD8⁺ T cell numbers in the lungs between these two groups and it is rather unlikely that this is responsible for the increased pathology. SFV alone does not cause histopathological changes in the lungs.

We have included histopathological assessment of lungs sections following SFV infection in the revised Figure 1d.

Fig. 1d Histopathological changes in the lungs of SFV, IAV, and SFV→IAV infected mice on 7 dpi. Lungs from naïve mice were included as control. Lung sections were stained with HE; representative images are shown. Bar = 200 μ m for x5; 100 μ m for x15; and 20 μ m for x40 magnification. Yellow arrows point to area of inflammation and damage. **e** Quantification of the extent of inflammation and damage in SFV, IAV, and SFV→IAV groups.

Minor Concerns

1.) *The conclusions regarding viral clearance are overstated. The data shows that on day 7 post infection neither group cleared the virus completely, therefore, they can only state at day 7 there was a higher viral load in the co-infected mice. They would need more timepoints to state if one group cleared the virus faster than the other. This should be reworded*

As suggested by the Reviewer, we have modified the sentence (page 5):

“It also leads to delayed clearance of IAV from the infected lungs” has been modified to read “It also leads to higher IAV titres in the lungs at a later stage of infection”.

2.) *Similarly, the cytokine data shows that there are still both tissue and circulating levels of cytokines in both the control and co-infected mice. While the levels of cytokine are higher in the co-infected mice, they are not absent in the control mice. The author should dampen the language that the cytokine storm is prolonged in the co-infected mice without showing data from more days.*

As suggested by the Reviewer, we have reworded the sentence (page 5) from:

“Thus, SFV infection predisposes to elevated and prolonged cytokine storm following subsequent IAV infection, both at the site of infection and systemically.”

To read:

“Thus, SFV infection predisposes to elevated cytokine storm following subsequent IAV infection, both at the site of infection and systemically.”

3.) *In the IAV trafficking to the brain section, this sentence is confusing: “After 3 hrs, mice were culled, the brain vasculature perfused, and brain dye levels determined. distributed throughout the body for 3 hr prior to culling mice and brain perfusion.”*

We thank the Reviewer for this comment. We agree, the second part of the sentence above has been inadvertently left in the submitted manuscript following final editing.

This has been modified in the revised manuscript and *distributed throughout the body for 3 hr prior to culling mice and brain perfusion* has been deleted.

The sentence now reads:

“After 3 hrs, mice were culled, the brain vasculature perfused, and brain dye levels determined.”

4.) *Did they use in vivo labelling to differentiate cells in the vasculature from cells in the tissue prior to identifying Trm? This is much more accurate than perfusion of the tissue.*

In vivo labelling to differentiate between cells in the blood and brain parenchyma was not performed in this study. We agree with the Reviewer that this method is more accurate and this has been demonstrated for the lung tissue. However, transcatheter perfusion is a gold standard when it comes to the brain perfusion and efficiently removes blood from the brain vasculature. In this study all brains were perfused prior to T cell isolation and identification of Trm cells. The comparison of intravascular antibody labelling and perfusion has shown that perfusion is efficient and sufficient to analyse Trm cell populations in the brain, including naïve brain (Ayasoufi *et al.* Brain, Behaviour, and Immunity, 2023, 112:51).

Following the Reviewer’s comment, we have discussed this in our revised version of the manuscript (page 19):

*The comparison of transcatheter perfusion with intravascular antibody labelling demonstrated that perfusion is sufficient to analyse cell populations in the brain, including naïve brain*⁶⁵.

5.) *Authors should state which clone of CD43 they used and if clone 1b11 is used it should be stated clearly that they are measuring changes in the glycosylation of CD43 rather than changes in CD43 expression.*

We thank the Reviewer for the attention to details. Clone S11 of CD43 was used in this study as part of the LEGENDScreen™ Mouse (PE) Kit (details available on the manufacturer’s website). S11 recognises CD43 regardless of glycosylation status (Baecher-Allan *et al.* Immunogenetics 1993;37:183).

6.) *The presence of CD8 T cells in the brain following IAV infection alone should be at least discussed.*

The presence of the CD8⁺ T cells in the brain of IAV infected mice is discussed on pages 6 and 16 of the submitted manuscript:

Conversely, of the very few CD8⁺ T cells infiltrating the brain in the IAV-only group, the majority (95.3%) were IAV tetramer-negative. We could minimally detect D^bNP₃₆₆⁺CD8⁺ T cells in 4 out of 15 mice and D^bPA₂₂₄⁺CD8⁺ T cells in 7/15 mice.

On the other hand, the majority of $CD38^+PD-1^+CD8^+$ T cells infiltrating the brain of IAV-only infected group were IAV tetramer-negative, suggesting recruitment of bystander $CD8^+$ T cells to the brain in the absence of SFV infection, presumably due to their inflammation-induced expansion in the periphery.

Since we demonstrated that i) IAV RNA is detectable in the brain of IAV-only infected mice (Figure 4d), ii) IAV infection affects BBB permeability (Figure 4a), and iii) IAV-specific $CD8^+$ T cells exhibit classical $CD25$ driven activation phenotype (Figure 2e), it is likely that IAV-specific $CD8^+$ T cells were recruited to the site of infection where cognate antigen was present. Conversely, tetramer-negative $CD8^+$ T cells detected in the brain of IAV-infected mice showed higher levels of KLRG1 and PD-1 expression (Figure 2e). These are likely to be recruited non-specifically, perhaps through the compromised BBB, and are likely to be short-lived effector cells arising as a result of bystander activation.

The above points were discussed on page 16 and 17 of the submitted manuscript.

7.) Fig 1b. What is the line connecting the 2 sets of dots curved? There is no data to support changes one way of the other between time points.

The graph in Figure 1b was generated using R package that plots the “line of best fit” between points. We have replaced the graph in Figure 1b with a bar graph showing individual points.

8.) Figure 2c. Numbers should be provided here along with proportions. Could this be complemented with a tetramer analysis?

Following the Reviewer’s comment, the numbers have been included in the revised Supplementary Material section (Supplementary Figure 1e).

9.) *The dot plots in figure 7c, 7f, and 8e are very difficult to interpret. The authors should not overlay the representative graphs.*

The dual-coloured dot plots in Figures 7 and 8 are concatenated plots representing the entire population of tetramer positive cells in the experimental group (n=5), not representative graphs, due to the relatively low numbers of cells. The two colours represent two different specificities tested relate to D^bNP₃₆₆ and D^bPA₂₂₄ tetramers. We feel that these plots combined with accompanied bar graphs are the best way to illustrate our findings.

As per Reviewer's suggestion, we have separated the specificities and plotted them as individual graphs. These data are included in the Supplementary Figure 4a-c of the revised manuscript.

Fig. S4a Concatenated FACS plots (n=5) of IAV-specific T_{EM} ($CD62L^{lo}CD44^{hi}$) and T_{CM} ($CD62L^{hi}CD44^{hi}$) of $D^bPA_{224}^+CD8^+$ T cell (left) and $D^bNP_{366}^+CD8^+$ T cell (right) specificities across different anatomical sites of IAV, SFV→IAV, and IAV→SFV infected mice at d90 post infection. **b** Concatenated FACS plots (n=5) of IAV-specific T_{RM} ($CD103^+CD69^+$) of $D^bPA_{224}^+CD8^+$ T cell (left) and $D^bNP_{366}^+CD8^+$ T cell (right) specificities in the lungs and brain of IAV, SFV→IAV, and IAV→SFV infected mice at d90 post infection. **c** Concatenated FACS plots (n=5) of IAV-specific $CD8^+$ effector T cells ($CD62L^{lo}CD44^{hi}$) for $D^bNP_{366}^+CD8^+$ and $D^bPA_{224}^+CD8^+$ T cells across different anatomical sites of IAV, SFV→IAV, and IAV→SFV PR8 challenged mice.

10.) Some of the discussion is overstated and needs to be revised.

Following the Reviewer's comment, we have read the Discussion thoroughly and "toned down" some of the statements that could have been construed as overstatements, eg. *uncontrolled inflammatory responses* was changed to *dysregulated inflammatory responses*, *greatly affects magnitude* to *affects magnitude*, *significantly slower resolution of viral infection* to *significantly higher viral infection*.

Reviewer: 3

This is an important question, however, the study design and investigation are limited and skewed to look at the question in a forward manner, without consideration of how the 2nd infection may impact the 1st and potentially contribute to the immunological changes observed. The inclusion of SFV, which is rarely found to infect humans and when found, tends to be much milder than in rodents, is not clear from a translational perspective. The authors should provide some rationale for the inclusion of SFV, rather than a different neurotropic virus that is translatable to a potential human condition.

We thank the Reviewer for the feedback and appreciation of our study.

SFV is a well-studied tractable model of experimental viral encephalitis¹⁰. In mice, the virus is 100% neuroinvasive following intraperitoneal injection and does not require intracranial inoculation as is the case with many other encephalitic viruses. SFV easily infects common strains of laboratory mice, and rodents may be its natural host. This negates the need for IFNAR^{-/-} mice used to study some other neurotropic viruses, for example flaviviruses¹¹. SFV infection in mice has a considerable body of supporting previous research and provides a well-regarded experimental system to study the pathogenesis of viral encephalitis.

We have added the above description to the revised manuscript on page 4 and adjusted the following paragraph that now reads:

SFV is a well-studied tractable model of experimental viral encephalitis¹⁰. In mice, the virus is 100% neuroinvasive following intraperitoneal injection and does not require intracranial inoculation as is the case with many other encephalitic viruses. SFV easily infects common strains of laboratory mice, and rodents may be its natural host. This negates the need for IFNAR^{-/-} mice used to study some other neurotropic viruses, for example flaviviruses¹¹. SFV infection in mice has a considerable body of supporting previous research and provides a well-regarded experimental system to study the pathogenesis of viral encephalitis. Adult C57BL/6 mice were infected with either 10⁴ pfu A/HKx31 (x31) IAV alone via the intranasal (i.n.) route, leading to localised respiratory infection; or 5x10³ pfu A7(74) SFV via the intraperitoneal (i.p.) route, which establishes systemic infection prior to neuroinvasion¹². In a third group, SFV infection was followed by i.n. IAV infection on day 8 post-SFV (SFV→IAV sequential infection) (**Figure 1a**). Day 8 was chosen as adaptive immune responses are at their peak and infectious SFV is generally cleared from the brain around this time^{13, 14}.

The absence of an SFV only group is disappointing. As the authors note, i.p. inoculation of SFV results in systemic infection. The potential for extracerebral SFV infection has been reported with i.p. inoculation and should be included in this study.

Please see responses to the Reviewer 2 point 1. SFV only control groups were included in the revised manuscript (Figure 1 and Figure 2 as well as Supplementary figure 1).

Figure 1 Foo *et al*

Figure S1 Foo *et al*

The potential for IAV exacerbating SFV injury and/or virus replication should also be investigated, as it is reasonable to anticipate the SFV is not fully cleared at the time of IAV exposure.

We agree with the Reviewer that it is reasonable to anticipate ramifications for SFV infection arising from IAV infection. This forms a part of a separate project in our laboratory that focuses on SFV responses. In the current study, we investigated changes to anti-IAV immune responses and effects of prior SFV infection.

We added this point to the revised manuscript on page 15 in the Discussion section.

While this study focused on responses to IAV, it is reasonable to anticipate perturbed immune responses to SFV following IAV infection. These investigations are currently ongoing.

According to Figure 6, it does appear that SFV is in lung at day 3 post-IAV infection but is unclear how long this persists. It is reasonable to expect that SFV does not get cleared and is presumably still present in the CNS.

Figure 6a shows SFV titres in the lung following SFV infection only. We have revised Figure 6a and added additional timepoints, showing that infectious SFV can no longer be detected in lungs on day 5 and 7 post-infection. We have not titrated SFV in the lungs in SFV→IAV group, since at the time of IAV co-infection (day 8 post-SFV infection), infectious SFV is only detectable in the brain. This has been shown in our previous studies (Fragkoudis *et al.* *Viruses*, 2018, 10:273, Kedzierski *et al.* *Viruses*, 2022, 14:2476).

We have also modified the sentence on page 12 of the revised manuscript to read:

SFV could be detected in the lungs on day 3 post-infection, and we also detected the presence of SFV-specific CD8⁺ T cells in the lungs of mice exposed to SFV or SFV→IAV (Figure 6a).

Pathology in all major organ systems should be investigated through histopathological techniques, including thorough investigation into organ distribution of both viruses and cell-type analyses. Due to the route of SFV infection, gut is a significant system that should also be included.

Reviewer raises good points here, which are reasonable questions, but taken together these would be a considerable body of work which would add knowledge, but which would not materially affect our novel finding that immune responses in the lungs to the influenza virus are perturbed by a prior SFV neurotropic infection.

Distribution of SFV in different tissues has been investigated previously using a high dose (10^8 pfu as opposed to 5×10^3 pfu used in our study) of luciferase-reporter SFV (Rodriguez-Madoz *et al.* Mol Therapy, 2007, 15:2164). Results showed that SFV could be detected in the gut following either i.v. or i.p. inoculation up to day 3 post infection but luciferase activity (hence SFV presence) decreased rapidly with time. In addition, infection pattern of virulent and avirulent SFV strains in mice demonstrated that both strains were detected in spleen, hind limb muscles, and aorta from 24h to 72h post-infection (Pusztai *et al.* British journal of experimental pathology, 1971, 669-677, 52(6)). This is reminiscent of our data in lungs shown in Figure 6a.

IAV infection is at 8-days post-SFV infection, which has not provided sufficient time for the mice to even partially recover from CNS injury caused by SFV. As such, it is not surprising that IAV-specific CD8⁺T cells would be present in the CNS, which are 'called-in' by CNS-mediate chemokines because of the on-going inflammation in the brain. This really isn't a 'redirection', as suggested by the authors, but rather the immunological consequence of two injured organ systems eliciting T-cell help.

We agree with the Reviewer that chemokine gradient in the CNS may have facilitated trafficking of IAV-specific CD8⁺ T cells to the brain, however, our data demonstrated that OT-I cells trafficked to the brain of mice infected with a virus carrying OVA peptide (SFV→IAV-OVA), but not in mice that were infected with the wild type virus (SFV→IAV). This finding further suggests that trafficking of IAV-specific CD8⁺ T cells to the brain is specific and antigen driven.

We have removed the word “redirect” in the modified manuscript (page 4 and 16) as per Reviewer’s suggestion.

The brain should be investigated for pathology and persistent SFV, particularly since only a subset of animals were demonstrated to have IAV-specific CD8 T-cells, which may suggest some animals cleared SFV or have less severe CNS pathology than others. Importantly, did any SFV infected animals experience encephalitis? Please provide the H&E images of the brain that are described in the Methods section.

The Methods section from an earlier version of the manuscript containing description of brain histopathology has been inadvertently left in the submitted version of the manuscript. This has been modified in the revised version of the manuscript (page 24).

Nevertheless, the brain pathology has been investigated in all experimental groups. It is important to note that mice used for histopathology examination were different to those used for immunological assays. Also, infectious SFV does not persist in the brain, but viral RNA persists for life of the animal (Fragkoudis *et al.* Viruses, 2018, 10:273).

In our experience all mice infected with SFV develop encephalitis, but the extent of symptoms varies between animals, and some might clear virus earlier than others. We have included a histopathology figure here (see below) showing that in this study, SFV-infected mice develop meningoencephalitis typical of SFV infection and characterised by neuronal necrosis, meningeal infiltration, diffused gliosis and vacuolation throughout different parts of the brain.

The Methods section states that SFV in brain was assessed with Vero cells, however, this is only shown in lung.

The Methods section was modified in the revised version of the manuscript to correctly reflect the manuscript content.

Please include tissue thickness in histopathological methods within the Methods section.

As requested by the Reviewer, we included the following information on page 24 in the Methods section: *Multiple sections were cut through the lungs at 5 μm thickness at 100 μm intervals (x5 levels).*

Sagittal H&E stained brain section from mice infected with SFV, IAV or SFV→IAV compared to naïve brain. SFV infected mice display meningoencephalitis changes characteristic of SFV infection: perivascular cuffing, meningeal infiltration or microgliosis and vacuolation in cerebellum. Some of the symptoms are still apparent in SFV→IAV mice, although they seem to be receding presumably due to the viral clearance from the brain (day 15 post-SFV infection).

REVIEWERS' COMMENTS

Reviewer #1 (Remarks to the Author):

Dear editor dear authors,

The authors responded to my comments in a vigorous and adequate manner. I consider this highly interesting manuscript ready for publication.

Kind regards

Bert Schepens

Reviewer #2 (Remarks to the Author):

The authors have addressed my concerns

Reviewer #3 (Remarks to the Author):

The authors have adequately addressed the concerns raised. I have no other concerns.